# Graph Assisted Offline-Online Deep Reinforcement Learning for Dynamic Workflow Scheduling

**Yifan Yang**[1], **Gang Chen**[1], **Hui Ma**[1], **Cong Zhang**[2,*], **Zhiguang Cao**[3], **Mengjie Zhang**[1]
[1]Victoria University of Wellington, [2]Nanyang Technological University,
[3]Singapore Management University
{yifan.yang, gang.chen, hui.ma, mengjie.zhang}@ecs.vuw.ac.nz
cong030@e.ntu.edu.sg, zgcao@smu.edu.sg

## Abstract

Dynamic workflow scheduling (DWS) in cloud computing presents substantial challenges due to heterogeneous machine configurations, unpredictable workflow arrivals/patterns, and constantly evolving environments. However, existing research often assumes homogeneous setups and static conditions, limiting flexibility and adaptability in real-world scenarios. In this paper, we propose a novel *Graph assisted Offline-Online Deep Reinforcement Learning* (GOODRL) approach to building an effective and efficient scheduling agent for DWS. Our approach features three key innovations: (1) a *task-specific* graph representation and a *Graph Attention Actor Network* that enable the agent to dynamically assign focused tasks to heterogeneous machines while explicitly considering the future impact of each machine on these tasks; (2) a *system-oriented* graph representation and a *Graph Attention Critic Network* that facilitate efficient processing of new information and understanding its impact on the current state, crucial for managing unpredictable workflow arrivals/patterns in real-time; and (3) an *offline-online* method that utilizes imitation learning for effective offline training and applies gradient control and decoupled high-frequency critic training techniques during online learning to sustain the agent's robust performance in rapidly changing environments. Experimental results demonstrate that GOODRL significantly outperforms several state-of-the-art algorithms, achieving substantially lower mean flowtime and high adaptability in various online and offline scenarios.

## 1 Introduction

The advent of cloud computing has revolutionized the way computational resources are utilized and managed, enabling organizations to execute complex *workflows* efficiently (Jadeja & Modi, 2012; Marinescu, 2022). Each workflow is typically represented as a Directed Acyclic Graph (DAG) (see Appendix A), where nodes correspond to *tasks* and edges represent task dependencies (Deelman et al., 2015). The goal of Dynamic Workflow Scheduling (DWS) is to assign tasks to heterogeneous machines to minimize the *mean flowtime* across a long operating duration (Huang et al., 2022).

DWS is characterized by flexible task assignment across heterogeneous machines, unpredictable workflow arrivals and patterns, and rapidly changing environments, making it one of the most challenging scheduling problems studied in the literature (Jayanetti et al., 2024). Despite its practical importance, DWS has received comparatively less attention previously. In fact, existing methods often oversimplify or fail to effectively address all these dynamic aspects.

Current approaches to DWS mainly rely on static scheduling heuristics that ignore the complexity of real-time cloud environments. Hand-crafted heuristics, such as priority dispatching rules (PDRs) (Topcuoglu et al., 2002; Pham & Fahringer, 2020), are valued for their speed, intuitiveness, and ease of implementation, but require extensive expertise and time-consuming adjustments. In contrast,

---

*corresponding author

Genetic Programming-based Hyper-Heuristic (GPHH) can automatically design PDRs through iterative evaluation-and-evolution, making it a state-of-the-art approach for DWS (Xu et al., 2023; Chen et al., 2024). However, GPHH can be highly sensitive in performance to specific problem configurations (see Appendix K) and is unsuitable for online applications due to high instability.

Recent advancements in *Learning to Optimize* (L2O) research (Kool et al., 2019; Wu et al., 2022) have successfully demonstrated the potential of using reinforcement learning (RL) algorithms to train neural network-based PDRs (a.k.a., scheduling agents) to tackle static scheduling problems (Zhang et al., 2020; Song et al., 2022; Zhang et al., 2024). Relevant studies clearly indicate that proper state representations are crucial for DRL (deep RL) algorithms to learn effectively (Zhang et al., 2024). However, existing vector- or matrix-based state representations (Huang et al., 2022; Jayanetti et al., 2024; Zhu et al., 2024) cannot accurately capture sophisticated interactions between tasks and machines in the cloud (verified in Table 1). Although Graph Neural Networks (GNNs) can effectively process complex graph-based states (Sun & Yang, 2023; Grinsztajn et al., 2023; Zhang et al., 2024), their fixed graph structures, shared feature embeddings, and unmodified RL methods limit scalability for tackling large and highly dynamic cloud workflow scheduling problems. This calls for developing novel graph representations and neural network architectures that can properly manage and process the ever-changing state information, along with advanced learning methods to boost the adaptability and stability of the scheduling agents in both offline and online scenarios.

In this paper, we propose a novel *Graph assisted Offline-Online Deep Reinforcement Learning* (GOODRL) approach for learning a *scheduling agent* to tackle DWS, featuring three innovations: (1) We develop a *task-specific* graph and a Graph Attention Actor Network to facilitate flexible task assignment across heterogeneous machines. They together explicitly capture the future impact of each machine on the focused task at both topological and feature levels, significantly enhancing the agent's ability to precisely differentiate all eligible actions. (2) We design a *system-oriented* graph and a Graph Attention Critic Network to seamlessly integrate newly arriving workflows with existing ones. By enabling bi-directional information flow and self-attention across all task nodes in the graph, the scheduling agent can accurately respond to real-time changes in the system state. (3) We propose an offline-online RL method that combines *offline* imitation learning with *online* gradient control and decoupled high-frequency critic training techniques to sustain the agent's robust performance in rapidly changing environments. Experiments across diverse scenarios demonstrate the effectiveness and reliability of GOODRL. Our method tackles significantly larger and more dynamic scheduling problems, which have been under-explored in prior works, consistently reducing the mean flowtime compared to multiple state-of-the-art algorithms.

## 2 RELATED WORK

To motivate our research on learning-based approaches for DWS, we review prominent research works on relevant scheduling problems from recent years, focusing on graph representations, neural network architecture designs, and training methods. (See Appendix S for more details)

**Graph Representations.** Scheduling research typically uses disjunctive graphs (Song et al., 2022; Su et al., 2023; Corsini et al., 2024; Zhang et al., 2024) or Directed Acyclic Graphs (DAGs) (Mao et al., 2019; Luo et al., 2021; Sun et al., 2021; Zhu et al., 2024) to represent system states, capturing various information related to workflows/jobs and machines. While disjunctive graphs model static job shop scheduling (JSS) through single static graphs of all known jobs on homogeneous machines (Zhang et al., 2020; Song et al., 2022; Su et al., 2023), they fail to handle the dynamic nature of unpredictable workflows and heterogeneous machines in DWS. In cloud computing, each workflow is typically represented as a separate DAG, which captures task-related information only, ignoring machine-related information. Current research processes each static DAG individually (Mao et al., 2019; Zhu et al., 2024), failing to model the real-time interactions across multiple dynamically arriving workflows and preventing them from being used to solve DWS problems effectively.

To address this, we propose a novel real-time graph representation to comprehensively model the state of multiple workflows and heterogeneous machines. Unlike prior works where actor and critic networks rely on the same graph representation (Zhang et al., 2020; Song et al., 2022; Zhu et al., 2024), we design two distinctive graph representations: one for the actor (for action selection), and another for the critic (for value estimation). This allows the actor to focus on task-specific information while ensuring the critic has access to the overall operating status of the DWS system.

**Architecture Design.** Graph Neural Networks such as Graph Convolutional Network (GCN) (Kipf & Welling, 2017; Mao et al., 2019; Ni et al., 2021; Zhu et al., 2024), Graph Isomorphism Network (GIN) (Xu et al., 2018; Su et al., 2023), Heterogeneous GNN (Song et al., 2022; Wang & Gombolay, 2022), and Graph Attention Network (GAT) (Veličković et al., 2018; Song et al., 2022; Wang et al., 2023) are commonly used to process graph-based state representations. For example, Ni et al. (2021) used GCN for extracting node embedding followed by self-attention layers to jointly process multiple sub-graphs associated with different process stages. Zhang et al. (2020) utilized GIN to obtain node embeddings, which are concatenated through mean pooling and are further processed by the actor network. While effective, this approach can make it more challenging for the actor to distinguish between different actions, particularly in large graphs with many nodes. In view of this, Zhang et al. (2024) introduced two separate GAT networks in the actor to extract topological embedding and context-aware embedding individually.

To better capture workflow dynamics and accurately evaluate the current policy, we design separate architectures for the actor and critic networks respectively, allowing each to specialize in different aspects of the scheduling process.

**Training Methods.** Current learning-based methods for scheduling problems include RL (Su et al., 2023; Lei et al., 2023; Zhang et al., 2023), as well as supervised (Pan et al., 2023; Luo et al., 2023), unsupervised, and self-supervised learning techniques (Corsini et al., 2024; Pirnay & Grimm, 2024). Supervised learning relies on expensive labeled solutions (Corsini et al., 2024). Self-supervised learning is highly sensitive to sample quality (Pirnay & Grimm, 2024). RL-based methods are more suitable for dynamic scheduling in online settings, thanks to their ability to continuously learn and make real-time decisions in a changing environment. Existing RL methods include *value*-based (e.g., DDQN (Liu et al., 2022; Lei et al., 2023)), *policy*-based (e.g., REINFORCE (Mao et al., 2019; Zhang et al., 2024), ES-RL (Su et al., 2023)), and *Actor-Critic*-based (e.g., PPO (Lei et al., 2023; Zhang et al., 2023)) algorithms.

Different from existing studies that focus mainly on offline learning with limited adaptability, we propose a two-stage RL approach for effective offline learning and stable online adaptation in DWS.

## 3 PROBLEM FORMULATION

A DWS problem instance consists of a set $\mathcal{W}$ of dynamically arriving workflows and a collection $\mathcal{M}$ of heterogeneous machines. A workflow $W_i \in \mathcal{W}$ is modelled as a DAG (see Appendix A), denoted by $W_i = (\mathcal{O}_{W_i}, \mathcal{C}_{W_i})$, with a node set $\mathcal{O}_{W_i} = \{O_{i1}, \ldots, O_{in_i}\}$ representing tasks and an edge set $\mathcal{C}_{W_i} = \{(O_{ij}, O_{ik}) | O_{ij}, O_{ik} \in \mathcal{O}_{W_i}\}$. Each edge $(O_{ij}, O_{ik})$ indicates that task $O_{ij}$ must be completed before $O_{ik}$ starts. Let $\mathcal{O} = \mathcal{O}_{W_1} \cup \mathcal{O}_{W_2} \cup \cdots \cup \mathcal{O}_{W_{|\mathcal{W}|}}$ be the set of all tasks to be executed to accomplish all workflows in $\mathcal{W}$. Each task $O_{ij}$ with a workload $tw_{ij} \in \mathbb{R}^+$ can be assigned to any machine $M_q \in \mathcal{M}$. Its execution time $et_{ij}^{(q)} = \frac{tw_{ij}}{ms_q}$ depends on the speed $ms_q$ of the assigned machine $M_q$. Each workflow $W_i$ arrives at a specific time $at_i$, and is finished at $ft_i = \max_j(ft_{ij})$ when all its tasks $O_{ij} \in \mathcal{O}_{W_i}$ are completed. The *flowtime* of workflow $W_i$ is determined by $F_i = ft_i - at_i$.

At each *decision step* $t$, a single **focused task** $O_t^*$ is identified. $O_t^*$ is an *unassigned task that becomes ready for execution at time* $t$ based on the current state of the DWS system. It will be assigned to a specific machine according to policy $\pi$. Once assigned, $O_t^*$ will be added to the machine's waiting queue. All pending tasks in the waiting queue are executed in the FIFO order (Senapati et al., 2021). The objective of DWS is to find a policy $\pi : M_q \sim \pi(O_{ij}, \mathcal{M})$ to minimize the mean flowtime $\bar{F} = \frac{1}{|\mathcal{W}|} \sum_{i=1}^{|\mathcal{W}|} F_i$ across all workflows received within an operating duration, while obeying task precedence constraints. Appendix B explains the workflow scheduling process in details.

## 4 METHODOLOGY

In this section, we introduce the proposed GOODRL approach. We first formulate DWS as a RL problem where our novel *task-specific* and *system-oriented* graph representations are introduced. Then, we detail the architecture designs of the GAT-based *actor* network and the GAT-based *critic* network. Finally, we outline the *offline-online* RL method.

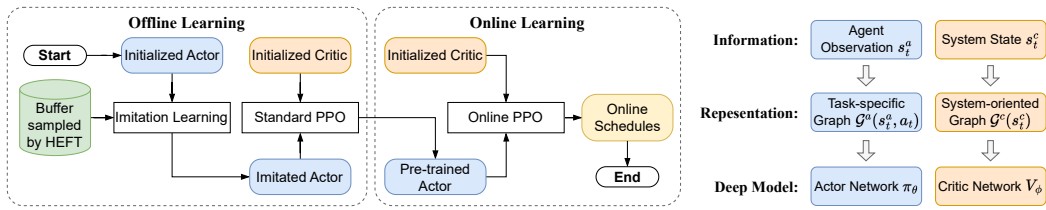

(a) The two-stage offline-online learning process.   (b) Information processed by the actor and critic.

Figure 1: **The overall framework of GOODRL for DWS.**

Figure 1 illustrates the overall framework of our GOODRL approach. In Figure 1(a), the *offline* stage begins by pre-training the actor network with imitation learning to follow expert-designed PDRs (e.g., HEFT (Topcuoglu et al., 2002; Senapati et al., 2021)). This prevents the accumulation of uncompleted tasks, which would otherwise occur if training started with a randomly initialized actor. Such accumulation can lead to memory issues and significantly increase computation time required for training the actor network. Standard PPO (Schulman et al., 2017) is subsequently employed to train the actor and critic networks. In the *online* stage, the pre-trained actor network is continuously fine-tuned by the newly developed online PPO algorithm. This is achieved by using gradient control and decoupled high-frequency critic training techniques introduced in Section 4.3.2. The information processed respectively by the actor and critic networks is shown in Figure 1(b).

## 4.1 REINFORCEMENT LEARNING PROBLEM FORMULATION

We model the scheduling process in the DWS system as an RL problem, as shown in Figure 2. The **Scheduling Agent** is an intelligent entity composed of an Actor, a Critic, and an RL Algorithm. As a model of policy $\pi$, the actor selects an action/machine $a_t$ to execute the focused task $O_t^*$ at any decision step $t$. The critic evaluates the actor's expected future return. The RL Algorithm continuously improves the performance of the actor under the guidance of the critic. The **Cloud Environment** employs a *Workflow Pool* to keep track of all workflows being executed on all available *Machine Resources*, providing system state $s_t^c$, agent observation $s_t^a$, and reward $r_t$ to the scheduling agent. The key components $(\mathcal{S}^c, \mathcal{S}^a, \mathcal{A}, \mathcal{P}, \mathcal{R}, \pi, V)$ of this RL problem are explained as follows.

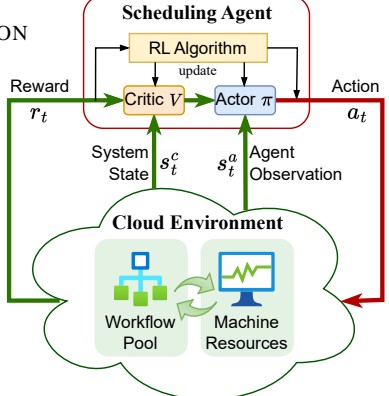

Figure 2: **The RL based formulation of the DWS systems.**

**System State.** Each system state $s_t^c \in \mathcal{S}^c$ is a snapshot of the current status of the whole DWS system at decision step $t$, including (1) the focused task $O_t^*$; (2) details of all workflows under processing, including information of their constituent tasks, task dependencies modeled by DAGs, as well as workflow arrival times; and (3) the current processing status of each machine. We design a novel **system-oriented graph** $\mathcal{G}^c(s_t^c) = (\mathcal{O}_t, \mathcal{C}_\mathcal{W} \cup \mathcal{C}_\mathcal{M} \cup \mathcal{C}_\mathcal{A})$ to represent $s_t^c$ as the input of the critic $V$, as shown in Figure 3(a). Only uncompleted tasks are included in $\mathcal{G}^c(s_t^c)$ for efficient memory usage.

Moreover, the edge set in $\mathcal{G}^c(s_t^c)$ covers three types of information: (1) *task precedence constraints* $\mathcal{C}_\mathcal{W}$ (e.g., edge $(O_{12}, O_{14}) \in \mathcal{C}_\mathcal{W}$ in Figure 3(a) indicates that task $O_{14}$ cannot be executed before completing task $O_{12}$); (2) *machine processing order constraints* $\mathcal{C}_\mathcal{M}$ (e.g., edge $(O_{13}, O_{23}) \in \mathcal{C}_\mathcal{M}$ in Figure 3(a) indicates that machine $M_1$ must execute $O_{13}$ first before executing $O_{23}$); and (3) the *eligible actions/machines for executing $O_t^*$* represented by $\mathcal{C}_\mathcal{A} = \{(O_{last}^{(a_t)}, O_t^*) \mid a_t \in \mathcal{A}\}$. Each edge in $\mathcal{C}_\mathcal{A}$ connects the last task node $O_{last}^{(a_t)}$, denoting the most recent task assigned to an eligible machine, to $O_t^*$ (e.g., task $O_{23}$ connected by edge $(O_{23}, O_{31}) \in \mathcal{C}_\mathcal{A}$ in Figure 3(a) is the last task assigned to machine $M_1$). In summary, $\mathcal{G}^c(s_t^c)$ models the evolving interactions across all workflows at each system state $s_t^c$, providing the critic with a comprehensive view of the DWS system.

**Agent Observation.** The agent observation $s_t^a$ at decision step $t$ is a partial observation of the system state $s_t^c$. To accurately capture subtle differences among all candidate actions for executing $O_t^*$, we propose a novel **task-specific graph** $\mathcal{G}^a(s_t^a, a_t) = (\mathcal{O}_t, \mathcal{C}_\mathcal{W} \cup \mathcal{C}_\mathcal{M} \cup \mathcal{C}_{a_t})$ to represent observation-action pair $(s_t^a, a_t)$ as the input of the actor $\pi$, as shown in Figure 3(b). The new graph representation explicitly captures the future impact of each specific action/machine $M_{a_t}$ on $O_t^*$ from two levels:

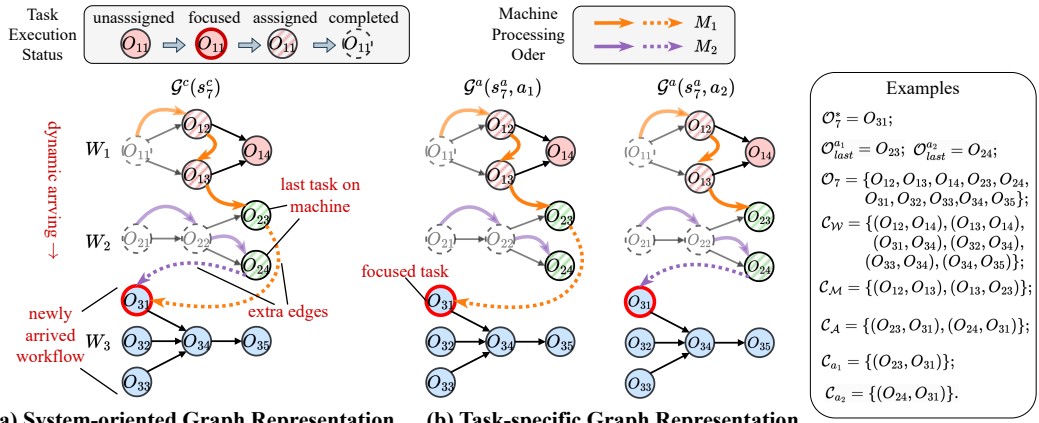

Figure 3: **Examples of two novel graph representations.** The DWS system at the current decision step includes 3 workflows and 2 machines, requiring the assignment of the focused task $O_{31}$ to either machine $M_1$ or $M_2$.

(1) At the *topology* level, an edge $\mathcal{C}_{a_t} = \{(O_{last}^{(a_t)}, O_t^*)\}$ is inserted in $\mathcal{G}^a(s_t^a, a_t)$ to capture the interaction between $O_t^*$ and the selected machine $M_{a_t}$ (e.g., edge $(O_{23}, O_{31})$ in Figure 3(b), since $O_{23}$ is the last task assigned to machine $M_1$, which is $M_{a_t}$ in this example). (2) At the *feature* level, the raw feature vector of $O_t^*$ is updated in real-time to predict the consequence of processing $O_t^*$ using $M_{a_t}$. The real-time updated feature vector $\mathbf{f}_{ij}^{(q)} \in \mathbb{R}^7$ includes seven important features defined in Appendix C, including task execution status $es_{ij}$, task workload $tw_{ij}$, remaining workload of associated workflow $rw_{ij}$, task execution time $et_{ij}^{(q)}$, task completion time $ct_{ij}^{(q)}$, machine speed $ms_q$, and machine utilization $mu_q$.

**Actions**. Any action $a_t \in \mathcal{A}$ assigns the focused task $O_t^*$ to the waiting queue of an eligible machine $M_q$. The size of the action space is $|\mathcal{A}| = |\mathcal{M}|$.

**Transition**. The system transits from state $s_t$ to state $s_{t+1}$ after taking action $a_t$, which assigns the focused task at decision step $t$ to a specific machine for execution. See Figure 8 in Appendix B.

**Rewards**. A reward $r_t$ provides a scalar feedback signal for taking action $a_t$ at system state $s_t^c$. In line with the objective to minimize the mean flowtime $\bar{F}$, we define $r_t = -\sum_{W_i \in \mathcal{W}^c} F_i$, where $\mathcal{W}^c$ refers to the set of workflows completed in between decision steps $t$ and $t+1$. If no workflows are completed during the two steps, $r_t = 0$.

**Learning Objective**. The scheduling agent aims to learn an optimal policy (modeled by the actor) to maximize the expected total rewards (estimated by the critic) from the long-term operation of the DWS system in both offline and online scenarios.

### 4.2 GRAPH ATTENTION ACTOR AND CRITIC NETWORKS

In the following, we introduce the architecture designs of the Graph Attention Actor Network $\pi_\theta(s_t^a, a_t)$ and the Graph Attention Critic Network $V_\phi(s_t^c)$, respectively.

#### 4.2.1 ACTOR NETWORK ARCHITECTURE DESIGN

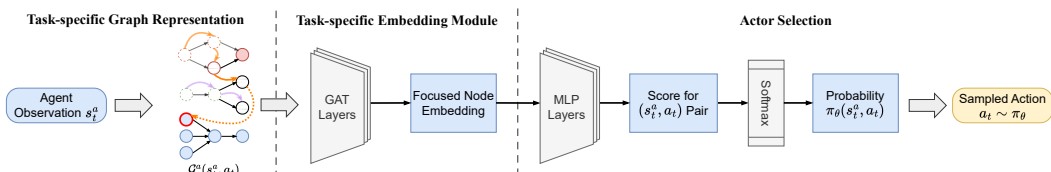

Figure 4: **Schematic design of the graph attention actor network.**

Figure 4 depicts the architecture design of the actor network $\pi_\theta(s_t^a, a_t)$ with trainable parameter $\theta$, aiming to accurately differentiate all eligible machines for processing the focused task. The actor network takes each observation-action pair represented by the task-specific graph $\mathcal{G}^a(s_t^a, a_t)$ as its input, and transforms $\mathcal{G}^a(s_t^a, a_t)$ into the node embedding of the focused task $O_t^*$ through *GAT layers*. By concentrating on the focused task/node, the subsequent *Multi-Layer Perceptron (MLP) layers* can easily produce discriminative scores for different $(s_t^a, a_t)$ pairs. These scores are further

processed by the *softmax* function to generate the probability distribution over the action space for reliable action selections by the actor.

**Task-specific Embedding Module.** In Figure 4, $K$ GAT layers (Veličković et al., 2018) are utilized to extract the node embedding of the focused task $O_t^*$ from $\mathcal{G}^a(s_t^a, a_t)$. Appendix D presents the detailed mathematical modelling of the GAT layers. The node embedding $\hat{\mathbf{h}}_{s_t^a, a_t} = \mathbf{h}_{O_t^*}^{(K)} \in \mathbb{R}^d$ with respect to the focused task $O_t^*$ at the $K$-th GAT layer is extracted from the task-specific graph $\mathcal{G}^a(s_t^a, a_t)$ and is further processed by an MLP for action selection.

**Action Selection.** To determine the probability of performing any action $a_t \in \mathcal{A}$ according to $\pi_\theta$, our actor network uses MLP layers to process the focused node embedding $\hat{\mathbf{h}}_{s_t^a, a_t}$ extracted from $\mathcal{G}^a(s_t^a, a_t)$. A score $z(s_t^a, a_t) = MLP_\theta(\hat{\mathbf{h}}_{s_t^a, a_t})$ is hence calculated. After obtaining the scores of all eligible actions, the action selection probability is determined using the popular softmax function: $\pi(s_t^a, a_t) = softmax(\{z(s_t^a, a_t)|a_t \in \mathcal{A}\})$. An action $a_t$ is then chosen following these probabilities.

**Remark.** Our actor network design introduces two advantages for flexible task assignment across heterogeneous machines. (1) *Pairwise processing*: Instead of processing all tasks and machines in a single static graph as in previous studies (Song et al., 2022; Su et al., 2023), we calculate each $(s_t^a, a_t)$ pair separately. This design explicitly considers the immediate and future impact of assigning any machine to the focused task at both the topology and feature levels, allowing the actor to accurately differentiate all eligible actions and make informed scheduling decisions. (2) *Focused embedding*: Our design learns the embedding of the focused task directly, rather than using mean pooling to combine embeddings of all nodes, as in many existing studies (Zhang et al., 2020; Corsini et al., 2024). This design can effectively prevent the learned embedding from being diluted by irrelevant information from non-focused tasks/nodes. Relevant ablation experiments on the actor network architecture design are reported in Appendix F.

### 4.2.2 CRITIC NETWORK ARCHITECTURE DESIGN

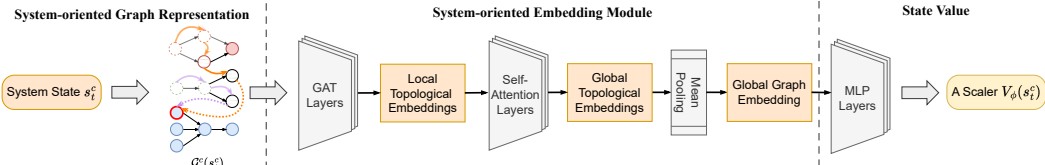

Figure 5: **Schematic design of the graph attention critic network.**

Figure 5 depicts the architecture design of the critic network $V_\phi(s_t^c)$ with trainable parameter $\phi$, aiming to capture the full information of the system state required for effective temporal-difference learning (Sutton, 2018). The critic network takes system-oriented graph representation $\mathcal{G}^c(s_t^c)$ as its state input. *GAT layers* are employed first by the critic to compute all node embeddings based on the local topological structure of $\mathcal{G}^c(s_t^c)$. Subsequently, *Self-Attention layers* are utilized to handle the relationship among all task nodes of $\mathcal{G}^c(s_t^c)$ at the global scale. The resulting node embeddings are then aggregated via *mean pooling* and processed by *MLP layers* to predict the expected future return of the DWS system.

**System-oriented Embedding Module.** Unlike previous approaches where the actor and critic networks share a common feature extractor (Ni et al., 2021; Song et al., 2022; Wang et al., 2023), our critic network uses its own GAT layers and self-attention layers (Vaswani, 2017; Ni et al., 2021) to extract node embeddings from $\mathcal{G}^c(s_t^c)$. Specifically, $K$ GAT layers are used to obtain the embeddings of all nodes in $\mathcal{G}^c(s_t^c)$. The corresponding mathematical modelling of these layers can be found in Appendix E.

After obtaining all node embeddings, we employ $L$ layers of self-attention to integrate information from all nodes in $\mathcal{G}^c(s_t^c)$. The update rule for the $l$-th self-attention layer is:

$$\mathbf{u}^{(l)} = \text{softmax}\left(\frac{\left(\mathbf{u}^{(l-1)}\mathbf{W}_Q^{(l)}\right)\left(\mathbf{u}^{(l-1)}\mathbf{W}_K^{(l)}\right)^\top}{\sqrt{d}}\right)\left(\mathbf{u}^{(l-1)}\mathbf{W}_V^{(l)}\right) \tag{1}$$

where $\mathbf{u}^{(0)} = [\mathbf{e}_x^{(K)}]_{x \in \mathcal{O}_t} \in \mathbb{R}^{|\mathcal{O}_t| \times d}$ is the collection of node embeddings produced by the GAT layers, $\mathbf{u}^{(l)} = [\mathbf{e}_x^{(K+l)}]_{x \in \mathcal{O}_t} \in \mathbb{R}^{|\mathcal{O}_t| \times d}$ is the collection of node embeddings produced by the $l$-th

self-attention layer, and $\mathbf{W}_Q^{(l)}, \mathbf{W}_K^{(l)}, \mathbf{W}_V^{(l)} \in \mathbb{R}^{d \times d}$ are the learnable weight matrices of the $l$-th self-attention layer. Finally, all node embeddings are aggregated using the common *mean pooling* technique to construct the high-level global embedding of $\mathcal{G}^c(s_t^c)$ as follows:

$$\bar{\mathbf{e}}_{s_t^c} = \frac{1}{|\mathcal{O}_t|} \sum_{x \in \mathcal{O}_t} \mathbf{e}_x^{(K+L)} \tag{2}$$

where $\mathbf{e}_x^{(K+L)} \in \mathbb{R}^d$ is the embedding of node $x$, corresponding to a row of $\mathbf{u}^{(L)}$.

**State Value.** The state value is computed by passing the aggregated global embedding $\bar{\mathbf{e}}_{s_t^c}$ through an MLP, yielding $V_\phi(s_t^c) = MLP_\phi(\bar{\mathbf{e}}_{s_t^c})$. This scalar output from the critic network estimates the expected future return from system state $s_t^c$ upon following the actor.

**Remark.** Our critic network design brings two key benefits for handling unpredictable workflow arrivals and patterns. (1) *Comprehensive context awareness*: The GAT layers process each edge of $\mathcal{G}^c(s_t^c)$ in both directions. We further add additional edges between the focused task and all eligible machines in $\mathcal{G}^c(s_t^c)$, enabling the critic to directly assess the influence of both past and future tasks on the focused task, providing a comprehensive contextual understanding of task-machine interactions. (2) *Long-range interaction modeling*: It is well-established in the scheduling literature that distant task nodes in a complex graph such as $\mathcal{G}^c(s_t^c)$ can strongly influence each other (Ni et al., 2021). We employ a self-attention mechanism to capture long-range dependencies across all task nodes, including those belonging to newly arrived workflows, empowering the critic with a holistic view of the system's dynamics. Relevant ablation experiments on the critic network architecture design are reported in Appendix G.

### 4.3 Offline-Online Reinforcement Learning

Two learning stages are involved in training the scheduling agent. In the *offline* stage, we pre-train the agent's actor and critic networks to establish effective initial policies for the DWS system. In the *online* stage, pre-trained networks are fine-tuned in real-time to adapt to the changing environment.

#### 4.3.1 Offline Learning

We employ both the imitation learning method (Barde et al., 2020) and the PPO algorithm (Schulman et al., 2017) to train the actor and critic networks. Initially, the actor network is trained through imitation learning, with the widely used HEFT heuristic serving as the teacher. (Topcuoglu et al., 2002). This approach avoids the unnecessary trial-and-error required for training a randomly initialized actor, which can cause unprocessed tasks to accumulate in the DWS system, increasing memory demands and risking program interruption. Therefore, the actor network can be quickly and effectively trained to achieve comparable performance as HEFT with high sample efficiency.

Afterwards, the PPO algorithm is utilized to train both the actor and critic networks. Iteratively, PPO samples $N$ independent trajectories to update both networks, further enhancing the effectiveness of the actor while keeping the critic accurate. See Appendix H.1 for the pseudo-code of offline learning.

#### 4.3.2 Online Learning

We develop an online version of the PPO algorithm that simultaneously uses the actor to schedule newly arriving workflows while continuously improving its performance during the day-to-day operation of the DWS system.

The standard PPO algorithm, which relies on multiple short trajectories for robust policy updates, is not suitable for online learning, where stable actor updates must be performed on a single long-lasting trajectory. We improve PPO with two techniques for online learning, enabling the actor to be reliably and quickly fine-tuned to meet changing scheduling demands.

First, we implement *Gradient Control* to stabilize actor training by regulating the gradient magnitude. Specifically, we adopt the following threshold to bound the L2 norm of the policy gradient vector used for training the actor network:

$$\nabla_\theta J = \begin{cases} \nabla_\theta J, & \text{if } \|\nabla_\theta J\|_2 \leq \mu_{\text{prev}} + \sigma_{\text{prev}} \text{ and } \|\nabla_\theta J\|_2 \leq \tau_0, \\ \mathbf{0}, & \text{otherwise,} \end{cases}$$

where $\nabla_\theta J$ stands for the policy gradient. $\mu_{\text{prev}}$ and $\sigma_{\text{prev}}$ are the mean L2 norm of $\nabla_\theta J$ and the corresponding standard deviation from the previous training epoch. $\tau_0$ bounds the maximum allowed L2 norm of $\nabla_\theta J$. Whenever the L2 norm of $\nabla_\theta J$ is too large, we avoid using it to train the actor network (i.e., $\nabla_\theta J$ is set to the zero vector). This prevents abrupt change of the actor network, ensuring stable learning in dynamic environments.

Second, we train the critic network independently and at a higher frequency than the actor, fully decoupling their training processes. This separation prevents interference between the two networks during training, providing the actor with more stable guidance from the critic's accurate value estimates. The pseudo-code of our online PPO algorithm is presented in Appendix H.2.

## 5 Experiments

This section evaluates GOODRL's performance, starting with the experimental setup and baselines, followed by offline and online comparisons, and ablation studies to assess some key components.

### 5.1 Experimental Setup

**Environment Settings.** We follow the framework of Huang et al. (2022) to perform training and testing on distinct offline and online scenarios listed in Table 1 and Table 2. Each scenario involves multiple machines and a series of dynamically arriving workflows. A scenario contains either $5 \times 5$ or $6 \times 4$ machines, representing the number of machine configurations and units per configuration, respectively (see Appendix I.3 for details). Four popularly studied workflow patterns (Deelman et al., 2015) (see Figure 9 and Table 6 in Appendix A) are used to randomly create 30 workflows for offline training and 1k, 3k, 5k workflows for offline evaluation. In online testing, we further assess PDRs and scheduling agents on large scenarios with 5k, 10k, and 20k workflows under demanding real-time conditions. These workflows arrive dynamically following a Poisson distribution, with an arrival rate of $\lambda = \{5.4, 9\}$ workflows per hour.

**Model Configurations.** Detailed information regarding the network architecture, normalization method, offline imitation learning, offline PPO, online PPO, hardware/software platform, and algorithm implementation is presented in Appendix J. Our code and data are publicly available at https://github.com/YifanYang1995/GOODRL.

**Baselines.** We compare GOODRL against three well-known PDRs designed manually by DWS experts: *Earliest Start Time* (EST), *Predict Earliest Finish Time* (PEFT), and *Heterogeneous Earliest Finish Time* (HEFT) (Topcuoglu et al., 2002; Senapati et al., 2021). We also employ the state-of-the-art *GPHH* method for DWS as a baseline to evolve high-performing PDRs through extensive evolutionary search. For GPHH, we use standard parameter settings from previous studies (Xu et al., 2023), perform 30 independent runs with different random seeds and report the best results obtained (details in Appendix K). Given the limited research on GNN-based state representations for DWS, we compare GOODRL with *ERL-DWS* (Shen et al., 2024), an advanced DRL method with an advanced transformer-based neural network architecture. Their average performance is evaluated using five random seeds.

### 5.2 Performance Comparison in Offline Scenarios

Table 1 presents the mean flowtime achieved by GOODRL and competing algorithms across 12 offline scenarios. Each scenario contains 30 problem instances of 1k, 3k or 5k workflows each. Since the test performance of the best scheduling heuristic evolved by GPHH varies hugely across different scenarios, we report the best test results achieved by the top three heuristics obtained from 30 independent runs (see Appendix K for details). Despite our best efforts, including adding imitation learning, ERL-DWS showed no significant improvement in test performance. We hence report its best available results in Table 1. Appendix P further reports the inference time of these methods.

GOODRL achieves consistently low mean flowtime in Table 1, with an average rank of 1.17 across all scenarios, outperforming all baselines. Compared to expert-designed PDRs (EST, PEFT, HEFT), GOODRL significantly reduces the mean flowtime, with Gap differences up to 289.98%. Although GPHH slightly outperforms "Ours-Offline" (i.e., trained offline with GOODRL) on scenarios $\langle 5 \times 5, 5.4, 1k \rangle$ and $\langle 5 \times 5, 5.4, 3k \rangle$, the Gap differences are merely 1.24% and 0.15%. However, GPHH

Table 1: **Performance comparison in offline scenarios.** "$\langle 6 \times 4, 9, 1k \rangle$": each instance in this scenario contains 1000 workflows arriving at a rate of 9 workflows per hour, which need to be assigned to machines with 6 different configurations, with 4 units per configuration. "Obj.": the *mean flowtime* over 30 instances. "Gap": the gap to the best "Obj" in each row. "**bold**": the best result in each scenario. "**blue bold**": the average ranking across all scenarios of the best approach.

| Scenarios | EST | | PEFT | | HEFT | | GPHH | | ERL-DWS | | Ours-Offline | |
|---|---|---|---|---|---|---|---|---|---|---|---|---|
| | Obj. | Gap | Obj. | Gap | Obj. | Gap | Obj. | Gap | Obj. | Gap | Obj. | Gap |
| $\langle 5 \times 5, 5.4, 1k \rangle$ | 1243.15 | 204.51% | 551.30 | 35.04% | 509.95 | 24.91% | **408.24** | **0.00%** | 1889.47 | 362.83% | 413.29 | 1.24% |
| $\langle 5 \times 5, 9, 1k \rangle$ | 1152.40 | 177.94% | 510.55 | 23.14% | 478.44 | 15.39% | 430.28 | 3.78% | 2180.41 | 425.89% | **414.61** | **0.00%** |
| $\langle 6 \times 4, 5.4, 1k \rangle$ | 1083.02 | 290.07% | 438.40 | 57.90% | 391.61 | 41.05% | 322.52 | 16.16% | 713.87 | 157.11% | **277.65** | **0.00%** |
| $\langle 6 \times 4, 9, 1k \rangle$ | 990.20 | 248.92% | 391.17 | 37.84% | 357.95 | 26.13% | 300.20 | 5.78% | 1523.83 | 436.95% | **283.79** | **0.00%** |
| $\langle 5 \times 5, 5.4, 3k \rangle$ | 1235.14 | 202.87% | 551.33 | 35.19% | 508.10 | 24.59% | **407.81** | **0.00%** | 2670.81 | 554.91% | 408.41 | 0.15% |
| $\langle 5 \times 5, 9, 3k \rangle$ | 1153.02 | 179.00% | 510.22 | 23.46% | 477.07 | 15.44% | 427.04 | 3.33% | 3582.70 | 766.91% | **413.27** | **0.00%** |
| $\langle 6 \times 4, 5.4, 3k \rangle$ | 1081.28 | 289.98% | 438.62 | 58.19% | 390.64 | 40.89% | 386.77 | 39.49% | 1108.95 | 299.96% | **277.27** | **0.00%** |
| $\langle 6 \times 4, 9, 3k \rangle$ | 992.46 | 250.72% | 389.94 | 37.80% | 356.08 | 25.83% | 358.40 | 26.65% | 2748.28 | 871.19% | **282.98** | **0.00%** |
| $\langle 5 \times 5, 5.4, 5k \rangle$ | 1231.70 | 202.34% | 550.53 | 35.13% | 507.91 | 24.67% | 408.38 | 0.24% | 2944.35 | 622.73% | **407.39** | **0.00%** |
| $\langle 5 \times 5, 9, 5k \rangle$ | 1146.62 | 177.17% | 509.61 | 23.19% | 477.12 | 15.33% | 427.88 | 3.43% | 4299.75 | 939.38% | **413.68** | **0.00%** |
| $\langle 6 \times 4, 5.4, 5k \rangle$ | 1076.75 | 288.11% | 437.53 | 57.71% | 389.24 | 40.30% | 386.95 | 39.47% | 1281.00 | 361.73% | **277.44** | **0.00%** |
| $\langle 6 \times 4, 9, 5k \rangle$ | 992.92 | 250.55% | 388.68 | 37.22% | 356.47 | 25.85% | 297.40 | 5.00% | 3480.87 | 1128.92% | **283.25** | **0.00%** |
| | 5.08 | | 4 | | 2.92 | | 1.92 | | 5.92 | | **1.17** | |

exhibits extensive performance variability, with Gaps reaching 39.49% and 39.47% to GOODRL in other scenarios. ERL-DWS performs significantly worse than GOODRL as high as 1128.92%. Additionally, both expert-designed PDRs and GOODRL exhibit robust performance as the number of workflows increase, while GPHH and ERL-DWS deteriorate significantly. Overall, our offline results demonstrate the impressive performance of GOODRL on DWS problems.

## 5.3 PERFORMANCE COMPARISON IN ONLINE SCENARIOS

Table 2 presents the mean flowtime obtained by all algorithms in online scenarios. "Ours-Offline" refers to the online performance of the scheduling agent only trained offline by GOODRL. "Ours-Online" indicates the online performance of the scheduling agent that is trained both offline and online by GOODRL. "Ours-Online" achieves the highest rank of 1.17 across all compared algorithms. Both "Ours-Online" and "Ours-Offline" significantly outperform expert-designed PDRs (EST, PEFT, HEFT) and ERL-DWS. While GPHH ranks third, it relies on the best result of 30 runs, requiring approximately 200 CPU hours for training. Additionally, GPHH's performance degrades as the number of workflows increases. In contrast, "Ours-Online" consistently improves upon "Ours-Offline", with performance gains of up to 1.24% in the $\langle 6 \times 4, 9, 20k \rangle$ scenario. This highlights GOODRL's ability to adapt pre-trained agents to dynamic demands (see Appendix Q for more robustness experiments), owing to its dynamic graph representation and offline-online training scheme. Note that even small improvement in mean flowtime can yield substantial practical benefits, including significant cost savings in large-scale dynamic environments (see Appendix R).

Figure 6 shows the *mean flowtime* achieved by "Ours-offline" and "Ours-online" in three online scenarios, comparing their performance over 5000 consecutive workflows. To ensure fair and reliable comparison, we begin tracking mean flowtime after the first 2000 workflows, by which point the DWS system has reached stable dynamics. "Ours-online" consistently achieves lower mean flowtime than "Ours-offline" throughout the entire evaluation, as clearly evidenced in Figure 6. These results highlight the effectiveness of online learning in GOODRL for continuously improving scheduling performance.

## 5.4 ABLATION STUDIES

(1) **Actor Network**: Table 4 (Appendix F) validates the task-specific embedding module (TSEM) in our actor network. Our-TSEM, with pairwise processing and focused task embedding, achieved the lowest cross-entropy loss compared to TSEM w/o pair (removing pairwise processing) and TSEM w. mean (adding mean pooling). These results confirm that separating task-machine pairs and avoiding mean pooling greatly enhance the ability to differentiate actions by focusing on critical task-specific information. (2) **Critic Network**: Table 5 (Appendix G) validates the system-oriented embedding module (SOEM) in our critic network. Ours-SOEM, incorporating bi-directional edges, additional connections, and a self-attention mechanism, significantly outperforms SOEM w/o. edge

Table 2: **Performance comparison in online scenarios.** "Scenarios": the scenarios for evaluating the online performance of the scheduling agent previously trained offline in $\langle 5 \times 5, 5.4 \rangle$. "**bold**": the best result in each scenario. "**blue bold**": the average ranking across all scenarios of the best approach.

| Scenarios | EST | | PEFT | | HEFT | | GPHH | | ERL-DWS | | Ours-Offline | | Ours-Online | |
|---|---|---|---|---|---|---|---|---|---|---|---|---|---|---|
| | Obj. | Gap | Obj. | Gap | Obj. | Gap | Obj. | Gap | Obj. | Gap | Obj. | Gap | Obj. | Gap |
| $\langle 6 \times 4, 5.4, 5k \rangle$ | 1076.01 | 277.05% | 439.28 | 53.93% | 391.63 | 37.23% | 303.70 | 6.42% | 1349.12 | 372.74% | 286.43 | 0.37% | **285.38** | **0.00%** |
| $\langle 6 \times 4, 5.4, 10k \rangle$ | 1077.09 | 279.13% | 439.64 | 54.75% | 390.26 | 37.37% | 305.31 | 7.47% | 1778.26 | 525.94% | **284.09** | **0.00%** | 285.12 | 0.36% |
| $\langle 6 \times 4, 5.4, 20k \rangle$ | 1072.90 | 276.97% | 439.88 | 54.55% | 391.18 | 37.44% | 309.12 | 8.61% | 2257.78 | 693.29% | 286.08 | 0.52% | **284.61** | **0.00%** |
| $\langle 6 \times 4, 9, 5k \rangle$ | 994.00 | 233.40% | 387.84 | 30.09% | 355.51 | 19.24% | 303.57 | 1.82% | 1246.91 | 318.24% | 301.00 | 0.96% | **298.14** | **0.00%** |
| $\langle 6 \times 4, 9, 10k \rangle$ | 993.97 | 238.09% | 387.64 | 31.85% | 355.21 | 20.82% | 307.27 | 4.52% | 1838.20 | 525.24% | 297.19 | 1.09% | **294.00** | **0.00%** |
| $\langle 6 \times 4, 9, 20k \rangle$ | 997.53 | 231.28% | 388.79 | 29.12% | 356.39 | 18.36% | 312.56 | 5.08% | 2783.78 | 835.93% | 301.11 | 1.24% | **297.44** | **0.00%** |
| | 6 | | 5 | | 4 | | 3 | | 7 | | 1.83 | | **1.17** | |

Figure 6: **Performance comparison between "Ours-Offline"** (trained only offline with GOODRL) **and "Ours-Online"** (trained both offline and online with GOODRL) **in three online scenarios.**

(removing bi-directional and additional edges) and SOEM w/o. self (removing the self-attention layers) in value loss. The results highlight the importance of comprehensive context awareness and modeling long-range interactions for effectively managing dynamic workflows. (3) **Online Learning**: Table 13 (Appendix L) validates two key techniques proposed for online training. Ours-Online achieved superior online performance improvement compared to Online w/o. grad. (remove gradient control) and Online w/o. freq. (remove independent high-frequency critic updates). These results demonstrate the effectiveness of both techniques in stabilizing and enhancing online learning.

## 5.5 SCALABILITY, TRANSFERABILITY, AND EXTENSIBILITY OF GOODRL

(1) **Scalability to significant changes**: Experiments in Appendix M demonstrate that our model can effectively handle significant changes in workflow patterns, arrival rates, and machine configurations without retraining. (2) **Transferability to FJSS**: We applied GOODRL to FJSS problems studied in Song et al. (2022), a representative research work on FJSS, to validate its transferability. Results in Appendix N demonstrate that GOODRL, designed for complex large-scale DWS problems, can also performs competitively on other scheduling problems such as FJSS. (3) **Extensibility to multi-objective problems**: In Appendix O, we demonstrate empirically that GOODRL can support other practical objectives such as cost, beyond flowtime reduction, by modifying the reward function. In particular, the actor trained with the modified reward can achieve a desirable trade-off between flowtime and cost, with a slight increase in flowtime but substantial cost savings of up to 41%.

## 6 CONCLUSION AND FUTURE WORK

This paper proposes GOODRL, an offline-online DRL approach for DWS. GOODRL features three key technical innovations. First, it introduces a new task-specific graph representation and a Graph Attention Actor Network to assess the immediate and future impacts of assigning any machine to process focused tasks, enabling the scheduling agent to dynamically and effectively manage workflow execution. Second, GOODRL adopts a novel system-oriented graph representation and a Graph Attention Critic Network to model complex interactions across multiple workflows and machines, providing accurate value estimates from a holistic system-wide perspective. Third, GOODRL leverages offline imitation learning for efficient actor pre-training and an enhanced online PPO algorithm with gradient control and decoupled high-frequency critic training techniques for robust real-time adaptation. Experimental results confirm that GOODRL significantly outperforms state-of-the-art baselines in minimizing mean flowtime. In future work, we could explore unlimited machine configurations, constraint handling and multi-objective learning techniques to tackle constrained multi-objective scheduling problems in cloud computing and other complex computing paradigms.

## 7 ACKNOWLEDGMENT

The authors thank Zaixing Sun, Zhengxin Fang, and Jian Liang for valuable discussions. We wish to acknowledge the use of New Zealand eScience Infrastructure (NeSI) high performance computing (HPC) facilities (https://www.nesi.org.nz) and the Rāpoi HPC cluster (https://vuw-research-computing.github.io/raapoi-docs/). This research is partially supported by Grant VUW-FSRG-10114, administered by Victoria University of Wellington. This research is also supported by the National Research Foundation, Singapore under its AI Singapore Programme (AISG Award No. AISG3-RP-2022-031).

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

# A DIRECTED ACYCLIC GRAPH

A workflow $W_i$ can be represented as a Directed Acyclic Graph (DAG), denoted by $W_i = (\mathcal{O}_{W_i}, \mathcal{C}_{W_i})$, as shown in Figure 7. Here, $\mathcal{O}_{W_i} = \{O_{i1}, O_{i2}, \cdots, O_{in_i}\}$ is the set of nodes, where each node $O_{ij}$ represents a specific *task* of $W_i$ that needs to be executed. The edge set $\mathcal{C}_i$ represents the directed edges between tasks, where each edge $(O_{ij}, O_{ik})$ indicates a precedence constraint such that task $O_{ij}$ must be completed before task $O_{ik}$ can start. The execution of tasks in the workflow follows the topological order defined by the DAG. A task can be scheduled or executed only after all its predecessor tasks have been completed; such a task is referred as a *ready task*.

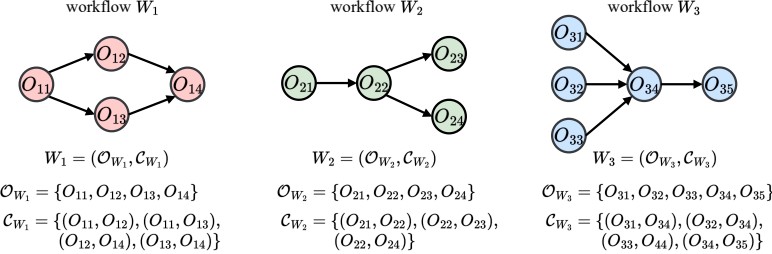

Figure 7: **Three workflows in DAG form.**

# B DYNAMIC WORKFLOW SCHEDULING PROCESS

Workflow scheduling involves assigning interdependent tasks to heterogeneous virtual machines to optimize specific objectives such as *mean flowtime* across workflows. In the scheduling process, certain constraints or assumptions must be taken into account:

- The pattern of each workflow (i.e., DAG structure) is unknown until it reaches the system.
- Tasks within a workflow can be allocated to any machine, with processing times varying according to machine speeds.
- Each machine can process only one task at a time.
- Only tasks with all predecessors completed are eligible for scheduling.

Figure 8 depicts the scheduling process in a DWS system. The process begins with the dynamic arrival of workflows into the *Workflow Pool*. At each decision step $t$, the system automatically identifies a unique task that needs to be assigned at state $s_t$, called the **focused task** $O_t^*$. For example, $O_{43}$ is chosen as the focused task at the current state. The scheduling agent then assigns the focused task to one of the available machine queues according to a *policy* $\pi$. For example, $O_{43}$ is assigned to the queue of machine $M_1$ following $M_1 = \pi(O_{43}, \{M_1, M_2, \cdots, M_{|\mathcal{M}|}\})$. The *assigned tasks* waiting in each machine queue will be executed according to the First-In-First-Out (FIFO) principle (Deelman et al., 2015; Huang et al., 2022), and their execution time depends on the machine's processing speed. The task execution status in the workflow pool is updated step-by-step, transitioning from **unassigned** to **focused**, then to **assigned**, and finally to **completed**. Meanwhile, the scheduling agent iteratively updates the system, identifies new focused tasks, and manages task execution in line with machine queues until all tasks are completed.

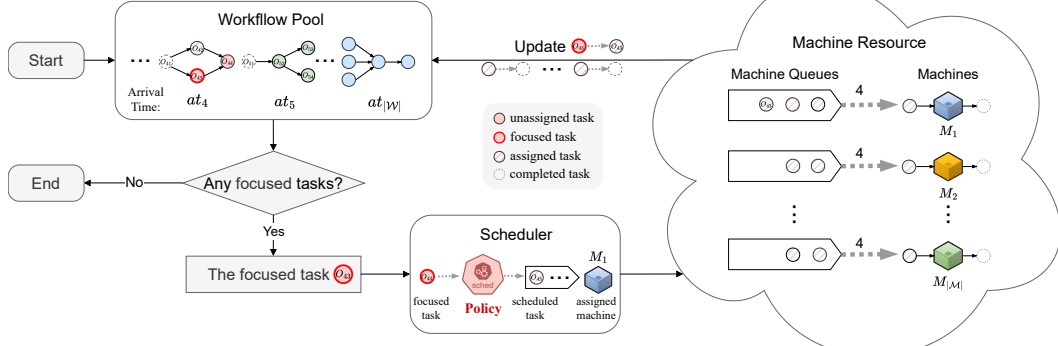

Figure 8: **Scheduling process in the dynamic workflow scheduling system.**

## C    RAW NODE FEATURES IN GRAPH REPRESENTATION

Detailed definitions of raw node features and their respective value ranges are reported in Table 3. In particular, for the four machine-related features (i.e., $et, ct, ms, mu$), if $\exists (O_{ij}, \cdot) \in \mathcal{C}_\mathcal{W} \cup \mathcal{C}_\mathcal{M} \cup \mathcal{C}_{a_t}$ or $\exists (\cdot, O_{ij}) \in \mathcal{C}_\mathcal{W} \cup \mathcal{C}_\mathcal{M} \cup \mathcal{C}_{a_t}$, the feature values $\mathbf{f}_{ij}^{(q)}$ of node $O_{ij}^{(q)}$ are calculated using the status of the specific machine $M_q$; otherwise, the feature value is *estimated* using the average information across all machines, denoted by $\mathbf{f}_{ij}^{(\bar{q})}$ on $M_{\bar{q}}$.

Table 3: **Meaning and value range of raw features within each node.**

|  | Symbol | Meaning | Explanation | Range |
|---|---|---|---|---|
| workflow related | $es_{ij}$ | Task execution status | Represents four possible status of task $O_{ij}$: unassigned, focused, assigned, and completed. | 0,1,2,3 |
|  | $tw_{ij}$ | Task workload | The workload of task $O_{ij}$. | $\leq 39185.28$ s |
|  | $rw_{ij}$ | Remaining workloads of associated workflow | The sum of the workloads of all unassigned tasks in workflow $W_i$ to which task $O_{ij}$ belongs. | $\leq 29.41$ h |
| machine related | $et_{ij}^{(q)}$ or $et_{ij}^{(\bar{q})}$ | Task Execution time | The execution time of task $O_{ij}$ on machine $M_q$, with $\bar{q}$ indicating the average across all machines. | $\leq 4898.16$s |
|  | $ct_{ij}^{(q)}$ or $ct_{ij}^{(\bar{q})}$ | Task completion time | The expected completion time of task $O_{ij}$ on machine $M_q$ or $M_{\bar{q}}$. | $\leq 29.41$ h |
|  | $ms_q$ or $ms_{\bar{q}}$ | Machine speed | The processing speed of machine $M_q$, with $\bar{q}$ representing the average processing speed across all machines. | 8,16,32,48,64,96 |
|  | $mu_q$ or $mu_{\bar{q}}$ | Machine utilization | the ratio of its working time to the total system time, with $\bar{q}$ representing the system average. | $[0,1)$ |

## D    TASK-SPECIFIC EMBEDDING MODULE

We use $K$ GAT layers (Veličković et al., 2018) to extract the node embedding of the focused task $O_t^*$ from $\mathcal{G}^a(s_t^a, a_t)$. Specifically, the update rule at the $k$-th layer is defined as follows:

$$\mathbf{h}_x^{(k)} = \sigma \left( \sum_{y \in \mathcal{N}(x)} \alpha_{xy}^{(k)} \mathbf{W}_\theta^{(k)} \mathbf{h}_y^{(k-1)} \right) \tag{3}$$

where $\mathbf{h}_x^{(0)} = \mathbf{f}_{ij}^{(q)} \in \mathbb{R}^7$ is the raw feature vector of node $x$ in $\mathcal{G}^a(s_t^a, a_t)$ at the input layer, and $\mathbf{h}_x^{(k)} \in \mathbb{R}^d$ is the $d$-dimensional embeddings of node $x$ at the $k$-th layer. Additionally, $\mathcal{N}(x) = \{y | \forall (y, x) \text{ or } \forall (x, y) \in \mathcal{C}_\mathcal{W} \cup \mathcal{C}_\mathcal{M} \cup \mathcal{C}_{a_t}\}$ is the neighboring set containing all incoming and outgoing neighbors of node $x$ in $\mathcal{G}^a(s_t^a, a_t)$. Furthermore, $\alpha_{xy}^{(k)}$ is the attention coefficient between nodes $x$ and $y$ at the $k$-th layer; $\mathbf{W}_\theta^{(k)}$ is the learnable weight matrix of the $k$-th layer; and $\sigma$ is a non-linear activation function (e.g., ReLU). The node embedding $\hat{\mathbf{h}}_{s_t^a, a_t} = \mathbf{h}_{O_t^*}^{(K)} \in \mathbb{R}^d$ with respect to the focused task $O_t^*$ at the $K$-th layer is extracted from the task-specific graph $\mathcal{G}^a(s_t^a, a_t)$ and is further processed by an MLP for action selection.

## E    SYSTEM-ORIENTED EMBEDDING MODULE

Unlike previous approaches where the actor and critic networks share a common feature extractor (Ni et al., 2021; Song et al., 2022; Wang et al., 2023), our critic network uses its own GAT layers and self-attention layers (Vaswani, 2017; Ni et al., 2021) to extract node embeddings from $\mathcal{G}^c(s_t^c)$. Specifically, $K$ GAT layers are used to obtain the embeddings of all nodes in $\mathcal{G}^c(s_t^c)$, with the update rule at the $k$-th layer defined as:

$$\mathbf{e}_x^{(k)} = \sigma \left( \sum_{y \in \mathcal{N}(x)} \alpha_{xy}^{(k)} \mathbf{W}_\phi^{(k)} \mathbf{e}_y^{(k-1)} \right) \tag{4}$$

where $\mathbf{e}_x^{(k)} \in \mathbb{R}^d$ is the $d$-dimensional embedding of node $x$ obtained at the $k$-th layer, $\mathbf{W}_\phi^{(k)}$ is the layer's learnable weight matrix, $\alpha_{xy}^{(k)}$ is the attention coefficient between nodes $x$ and $y$, and $\sigma$ is a non-linear activation function (e.g., ReLU). The neighborhood set $\mathcal{N}(x) = \{y | \exists (y, x) \text{ or } (x, y) \in \mathcal{C}_\mathcal{W} \cup \mathcal{C}_\mathcal{M} \cup \mathcal{C}_\mathcal{A}\}$ contains all predecessor and successor nodes connected directly to node $x$ in $\mathcal{G}^c(s_t^c)$.

Table 4: **Cross-entropy loss at different iterations.** "Ours-TSEM": our task-specific embedding module proposed in Section 4.2.1. "TSEM w/o. pair": an architecture that does not handle $(s, a)$ pairs; instead, it directly calculates all action selection probabilities based on a state. "TSEM w. mean": an architecture that additionally concatenates the mean pooling of all nodes from the task-specific graph as input to the MLP.

| Actor Architecture | 100-th | 200-th | 300-th | 400-th | 500-th | 600-th | 700-th | 800-th | 900-th |
|---|---|---|---|---|---|---|---|---|---|
| Ours-TSEM | 2.7486 | **2.7106** | **2.6881** | **2.6647** | **2.6498** | **2.6038** | **2.5726** | **2.5297** | **2.5091** |
| TSEM w/o. pair | 3.1707 | 3.1597 | 3.1538 | 3.1468 | 3.1435 | 3.1394 | 3.1365 | 3.1333 | 3.1302 |
| TSEM w. mean | **2.7099** | 2.7209 | 2.7152 | 2.6659 | 2.7109 | 2.6172 | 2.5989 | 2.5334 | 2.5243 |

## F  ABLATION STUDY OF ACTOR NETWORK ARCHITECTURE DESIGN

In Table 4, we compare three alternative actor network architectures based on their cross-entropy loss defined below averaged over five random seeds. This comparison provides insights into how well each architecture can learn from HEFT through imitation learning. The best architecture is expected to achieve the lowest cross-entropy loss and hence is more capable of modelling effective scheduling policies.

$$\mathcal{L}_{CE} = \frac{1}{|\mathcal{D}|} \sum_{s_t^a, a_t \in \mathcal{D}} \text{CrossEntropy}(\pi_\theta(s_t^a, \cdot), a_t) \tag{5}$$

where $\mathcal{D}$ is a repository of state-transition data obtained by using HEFT to schedule workflow tasks, $\pi_\theta(s_t^a, \cdot)$ represents the probability distribution over eligible actions determined by the actor network with parameters $\theta$.

We compare **Ours-TSEM**, which incorporates the advantages of *pairwise processing* and *focused embedding* mentioned in Section 4.2.1, against two baselines: **TSEM w/o pair** (removing pairwise processing) and **TSEM w. mean** (adding mean pooling over all nodes).

- **Our-TSEM** achieves the lowest cross-entropy loss in most cases, as its architecture precisely considers the impacts of each machine on the focused task, enabling the actor network to effectively differentiate all candidate actions.

- **TSEM w/o pair** demonstrates a smaller reduction in loss than Our-TSEM, indicating that *pairwise processing* can accurately capture the task-machine interactions, facilitating effective scheduling.

- **TSEM w. mean** leads to inferior loss, since it dilutes the information of the focused task as a result of mixing the embeddings of all tasks, making it hard to distinguish eligible actions.

Overall, Our design of TSEM allows precise differentiation of actions, facilitating precise task assignment across heterogeneous machines.

## G  ABLATION STUDY OF CRITIC NETWORK ARCHITECTURE DESIGN

In Table 5, we analyze the effectiveness of our critic network design based on the value loss, which measures the mean squared error between predicted state values and sampled returns:

$$\mathcal{L}_{MSE} = \frac{1}{|\mathcal{D}|} \sum_{s_t^c \in \mathcal{D}} (V_\phi(s_t^c) - R_t)^2, \tag{6}$$

where $V_\phi(s_t^c)$ represents the predicted value for state $s_t^c$ with network parameters $\phi$, and $R_t$ is the corresponding sampled return.

To validate the advantages of *comprehensive context awareness* and *long-range interaction modeling* in our critic network design proposed in Section 4.2.2, we compare **Ours-SOEM** against two variants: **SOEM w/o edge** (removing bi-directional and additional edges) and **SOEM w/o self** (removing the self-attention mechanism).

- **Ours-SOEM** achieves the lowest value loss in most cases, demonstrating the effectiveness of its design in capturing the full influence of task-machine interactions, enabling the critic to make accurate value estimations.

Table 5: **Mean relative error between return and state value at different iterations.** "Ours-SOEM": our system-oriented embedding module proposed in Section 4.2.2. "SOEM w/o. edge": only one-directional edges and no extra edges are used in Ours-SOEM. "SOEM w/o. self": Ours-SOEM architecture removes the self-attention layer.

| Critic Architecture | 100-th | 200-th | 300-th | 400-th | 500-th | 600-th | 700-th | 800-th | 900-th |
|---|---|---|---|---|---|---|---|---|---|
| Ours-SOEM | **16.3971** | 14.0938 | **10.4907** | **9.5811** | **7.8581** | 7.5675 | **7.1238** | **6.0035** | **6.2547** |
| SOEM w/o. edge | 17.3012 | **13.4737** | 11.6626 | 9.8066 | 8.8853 | **7.5266** | 7.5607 | 7.593 | 7.1468 |
| SOEM w/o. self | 20.6114 | 16.1826 | 14.6813 | 12.6997 | 12.0733 | 10.7019 | 10.1497 | 8.5121 | 8.5645 |

- **SOEM w/o edge** shows a higher value loss than Ours-SOEM, indicating that relying solely on unidirectional edges limits the critic's ability to capture the context of task-machine interactions. Without additional connections, it is challenging for **SOEM w/o edge** to capture comprehensive contextual relationships in the graph.

- **SOEM w/o self** exhibits a noticeable increase in value loss due to the lack of the self-attention layer, which is essential for modeling long-range interactions among task nodes. This capability is crucial for DWS, as tasks from newly arriving workflows can have a long-range impact on existing tasks.

Overall, our design of SOEM enhances contextual understanding and effectively captures long-range dependencies. For example, in Figure 3(a), the newly arrived workflow $W_3$ can be effectively incorporated into the current system state by using bi-directional and additional edges and the self-attention mechanism. Our design of SOEM enables the critic to provide accurate value estimations and adapt to the complexities of unpredictable workflow arrivals and patterns.

## H    DETAILS OF OFFLINE AND ONLINE LEARNING ALGORITHMS

### H.1    DETAILS OF OFFLINE LEARNING ALGORITHM

For offline learning, we propose a two-step training process that combines Imitation Learning and Proximal Policy Optimization (PPO) for offline training, as shown in Algorithm 1. In the first step, the actor leverages Imitation Learning to efficiently mimic the behavior of HEFT (Topcuoglu et al., 2002), an expert-designed heuristic policy denoted by $\hat{\pi}$, aiming to minimize the corresponding cross-entropy loss (*line 7*). This method allows the actor to quickly learn an effective policy, preventing the buildup of uncompleted tasks in the graph representation, which often occurs with randomly initialized policies.

The actor trained via imitation learning provides a stable starting point for the second step. In the second step, $N$=4 independent trajectories are collected per training iteration. The scheduling agent jointly updates both the actor and critic networks in the direction of minimizing the clipped policy loss and the value function loss (*line 24*). This combination of imitation learning followed by PPO ensures robust performance while mitigating the risk of memory overload.

### H.2    DETAILS OF ONLINE LEARNING ALGORITHM

For online learning, we propose an enhanced version of the Proximal Policy Optimization (PPO) algorithm, as outlined in Algorithm 2. It enables the actor to dynamically schedule incoming workflows while simultaneously refining its parameters during the operations of the Dynamic Workflow Scheduling (DWS) system. Traditional PPO relies on multiple short trajectories for effective policy updates, making it less suitable for online scenarios where stable updates are necessary based on a single long-lasting trajectory.

The algorithm alternates between real-time scheduling decisions (*line 3-6*) and network updates (*line 8-24*). The transition data by the interaction between the actor and the dynamic environment during real-time scheduling is efficiently managed and stored through a fixed-size experience buffer $\mathcal{B}$ (*line 6*). For dynamically arriving workflows, at each step, the system needs to sample an action/machine for the focused task based on the actor's policy. After every $T_w$ steps (after an initial warm-up period of $T_{nw}$), the most recent transitions in buffer $\mathcal{B}$ are used to update the actor and critic networks separately. The updated actor then resumes real-time decision-making, gathering new transition data for the next network update cycle.

---

**Algorithm 1:** Offline Learning with Imitation Learning and PPO

---

**Input:** Initial actor network $\pi_\theta$, initial critic network $V_\phi$, environment *env*, heuristic policy $\hat{\pi}$, actor learning rate $\alpha_\pi$, critic learning rate $\alpha_V$, value loss coefficient $c_v$, entropy loss coefficient $c_e$, clipping parameter $\epsilon$, number of trajectories $N$, update epochs $E_1$ and $E_2$, number of training steps $U$

**Output:** Pre-trained actor network $\pi_\theta$, pre-trained critic network $V_\phi$

   `// Step 1:  Imitation Learning`

1   Initialize environment *env* and buffer $\mathcal{D}$;

2   **while** *env is not terminated* **do**

3      Execute action $a_i = \hat{\pi}(s_t)$ and observe next state $s_{t+1}$;

4      Store transition $(s_t^a, a_t)$ in buffer $\mathcal{D}$;

5      $s_t \leftarrow s_{t+1}$;

6   **for** *update* = 1 *to* $E_1$ **do**

7      Compute cross-entropy loss: $\mathcal{L}^{CE} = \frac{1}{|\mathcal{D}|} \sum_{s_i^a, a_i \in \mathcal{D}} \text{CrossEntropy}(\pi_\theta(s_i^a, \cdot), a_i)$;

8      Perform gradient descent on $\mathcal{L}^{CE}$ to update $\theta$ to mimic heuristic policy $\hat{\pi}$;

   `// Step 2:  PPO Training`

9   **for** *each iteration* = 1 *to* $U$ **do**

10      Initial buffer $\mathcal{B}$; $\theta_{old} \leftarrow \theta$;

11      **for** *n* = 1 *to* $N$ **do**

12          Initialize environment $env_n$ and buffer $\mathcal{B}_n$;

13          **while** $env_n$ *is not terminated* **do**

14              Sample action $a_t \sim \pi_{\theta_{old}}(s_t)$;

15              Execute action $a_t$ and observe reward $r_t$ and next state $s_{t+1}$;

16              Store transition $(s_t^a, a_t, r_t, s_{t+1})$ in buffer $\mathcal{B}_n$;

17              $s_t \leftarrow s_{t+1}$;

18          **for** *each transition in buffer* $\mathcal{B}_n$ **do**

19              Calculate Return: $R_t = \sum_{k=0}^{|\mathcal{B}_n|-t} \gamma^k r_{t+k}$;

20              Calculate Advantage: $\hat{A}_t = R_t - V_\phi(s_t^c)$;

21          $\mathcal{B} \leftarrow \mathcal{B} \cup \mathcal{B}_n$;

22      **for** *update* = 1 *to* $E_2$ **do**

23          **for** *each minibatch* $\hat{\mathcal{B}}$ *from buffer* $\mathcal{B}$ **do**

24              Compute aggregated loss:

$$\mathcal{L}^{CLIP} = \frac{1}{|\hat{\mathcal{B}}|} \sum_{j=1}^{|\hat{\mathcal{B}}|} \min\left( \frac{\pi_\theta(a_j|s_j)}{\pi_{\theta_{old}}(a_j|s_j)} \hat{A}_j, \text{clip}\left( \frac{\pi_\theta(a_j|s_j)}{\pi_{\theta_{old}}(a_j|s_j)}, 1-\epsilon, 1+\epsilon \right) \hat{A}_j \right),$$
$$\mathcal{L}^{VF} = \frac{1}{|\hat{\mathcal{B}}|} \sum_{j=1}^{|\hat{\mathcal{B}}|} \left( V_\phi(s_j) - R_j \right)^2,$$
$$\mathcal{L} = -\mathcal{L}^{CLIP} + c_v \mathcal{L}^{VF} ;$$

25              Perform gradient descent on $\mathcal{L}$ to update $\theta$ and $\phi$;

---

One technique of our approach involves monitoring the gradient changes of the actor (*line 21-23*). This is performed by controlling the mean L2 norm of the gradients $\|\nabla J\|_2$ within a dynamically adjusted threshold (*line 21*). Actor updates are permitted only when the gradient norm remains below this threshold, facilitating stable online learning. Additionally, the upper bound on gradient changes is adaptively adjusted to maintain controlled updates, ensuring stability in policy improvement and mitigating abrupt changes in actor behavior. This stabilization is crucial in dynamic environments, where sudden shifts in actor's policy could lead to catastrophic scheduling decisions, resulting in severe backlogging of subsequent workflow tasks.

Another technique is the independent, higher-frequency training of the critic network (*line 11-14*). This decoupling from the actor's updates allows the critic to improve its accuracy of estimating future returns. Meanwhile, frequent critic update stabilizes online learning by providing more precise value estimations. Subsequently, the algorithm uses the updated critic to guide the training of the actor's parameter. Accurate critic predictions provide more reliable feedback for the actor, leading to more effective policy improvement.

---

**Algorithm 2:** Online Learning with Enhanced PPO

---

**Input:** Pre-trained actor network $\pi_\theta$, pre-trained critic network $V_\phi$, environment *env*, discounting factor $\gamma$, actor learning rate $\alpha_\pi$, critic learning rate $\alpha_V$, entropy loss coefficient $c_e$, clipping parameter $\epsilon$, environment warm-up steps $T_{nw}$, critic warm-up steps $T_{cw}$, time window steps $T_w$, actor update epochs $E_a$, critic update epochs $E_c$

**Output:** Scheduling decision of arrival workflows with $|\mathcal{O}|$ tasks

1 Initialize online environment *env*, and an experience buffer $\mathcal{B}$ with fixed size $2T_w$;

2 **for** $t = 1$ *to* $|\mathcal{O}|$ **do**

    `// Online Decision Making`

3     $\theta_{old} \leftarrow \theta$;

4     Sample action $a_t \sim \pi_{\theta_{old}}(s_t)$;

5     Execute action $a_t$ and observe reward $r_t$ and next state $s_{t+1}$;

6     Store transition $(s_t^a, a_t, r_t, s_{t+1})$ in buffer $\mathcal{B}$;

7     **if** $(t - T_{nw})\%T_w == 0$ *and* $t \geq T_{nw}$ **then**

8         **for** *each transition in buffer $\mathcal{B}$* **do**

9             Calculate Return using $T_w$ steps of rewards: $R_i = \sum_{k=0}^{T_w} \gamma^k r_{i+k}$;

10         **for** *update = 1 to $E_a$* **do**

            `// Update Critic Network`

11             **for** *update = 1 to $E_c/E_a$* **do**

12                 **for** *each minibatch $\hat{\mathcal{B}}$ from buffer $\mathcal{B}$* **do**

13                     Compute value loss: $\mathcal{L}_V = \frac{1}{|\hat{\mathcal{B}}|} \sum_{j=1}^{|\hat{\mathcal{B}}|} (V_\phi(s_j) - R_j)^2$;

14                     Perform gradient descent on $\mathcal{L}_V$ to update $\phi$;

            `// Update Actor Network`

15             **if** $(t - T_{nw})//T_w \geq T_{cw}$ **then**

16                 **for** *each transition in buffer $\mathcal{B}$* **do**

17                     Calculate Advantage: $\hat{A}_i = R_i - V_\phi(s_i)$;

18                 **for** *each minibatch $\hat{\mathcal{B}}$ from buffer $\mathcal{B}$* **do**

19                   Compute policy loss:

$$\mathcal{L}_\pi = \frac{1}{|\hat{\mathcal{B}}|} \sum_{j=1}^{|\hat{\mathcal{B}}|} \min\left(r_j(\theta)\hat{A}_j, \text{clip}\left(r_j(\theta), 1 - \epsilon, 1 + \epsilon\right)\hat{A}_j\right)$$

$$\text{where } r_j(\theta) = \frac{\pi_\theta(a_j|s_j)}{\pi_{\theta_{old}}(a_j|s_j)};$$

20                     Calculate the mean L2 norm of the actor's gradients $\|\nabla J\|_2$;

21                     **if** $\|\nabla J\|_2 \leq \mu_{prev} + \sigma_{prev}$ *and* $\|\nabla J\|_2 \leq \tau_0$ **then**

22                         Perform gradient ascent on $\mathcal{L}_\pi$ to update $\theta$;

23             Update the parameters of the gradient control $\mu_{\text{prev}}, \sigma_{\text{prev}}$;

24     $s_t \leftarrow s_{t+1}$;

---

# I SIMULATION SETTINGS

## I.1 REAL-WORLD TRACES

Regarding real-world traces, previous studies explored three different aspects, as summarized below:

- **Workflow characteristics**: Past studies considered different task dependency graphs, number of tasks, and task execution time. They focused mainly on scientific workflows such as CyberShake and Montage (Deelman et al., 2015; Qin et al., 2023; Sun et al., 2024). For a detailed introduction to these real-world scientific workflows, please refer to (Deelman et al., 2015). These datasets are available in the workflow repository on the Pegasus website (https://download.pegasus.isi.edu/misc/SyntheticWorkflows.tar.gz). In line with existing works, our experiments were also conducted on these scientific workflows.

- **Resource configurations**: Past studies considered various resource configurations, including virtual or physical machine configurations, quantities, and prices. They mainly followed re-

source configurations supported by major cloud providers, such as Amazon EC2[1], Google Cloud[2], and Microsoft Azure[3]. Our experiments adopted the resource configurations supported by Amazon EC2.

- **Arrival patterns**: Existing studies primarily relied on workflow arrivals simulated by a Poisson distribution, such as (Huang et al., 2022; Xu et al., 2023; Wang et al., 2019). Please refer to (Wang et al., 2019) for a theoretical analysis of such arrival patterns. Wang et al. (2019) specifically pointed out that "the distributions of the time interval of each request are exponential which implies that the arrival rates are Poisson distributed.".

## I.2 WORKFLOW INFORMATION

We adopt the common experimental setup used in the cloud computing domain (Huang et al., 2022; Xu et al., 2023), utilizing four widely-studied workflow patterns (Deelman et al., 2015), as shown in Figure 9. Each workflow pattern contains detailed information such as task workloads $tw_{ij}$ and the dependency relationships between tasks, as summarized in Table 6. For more information, please refer to https://pegasus.isi.edu/documentation/user-guide/introduction.html. In the simulated experiments, these workflow patterns arrive dynamically according to a Poisson distribution $\lambda$, consistent with the default settings of existing studies (Wang et al., 2019; Xu et al., 2023; Shen et al., 2024).

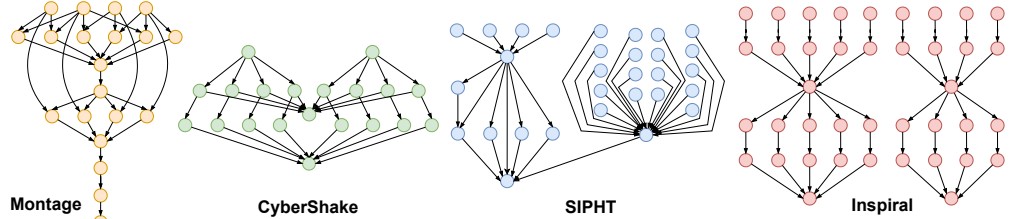

**Montage**    **CyberShake**    **SIPHT**    **Inspiral**

Figure 9: **Four widely studied workflow patterns in dynamic workflow scheduling.**

Table 6: **Information of four widely used workflow patterns.** "Total workload": the sum of all task workload $tw_{ij}$ in a workflow.

| Pattern name | Number of tasks | Number of edges | Average task workload | Total workload |
|---|---|---|---|---|
| Montage | 25 | 45 | 145.76 s | 1.01 h |
| CyberShake | 30 | 52 | 405.62 s | 3.38 h |
| SIPHT | 29 | 33 | 3060.12 s | 24.65 h |
| Inspiral | 30 | 35 | 3529.10 s | 29.41 h |

Figure 10 depicts the 95%-99% quantile of workflow arrivals at different arrival rates. Specifically, the 95% and 99% quantiles at $\lambda = 5.4$ are 9.3 (approx. 280 tasks) and 11.7 (approx. 350 tasks) workflows per hour, and that $\lambda = 9$ are 14 (approx. 420 tasks) and 16 (approx. 480 tasks) workflows per hour.

## I.3 MACHINE CONFIGURATIONS

The cloud system in our experiments supports six types of virtual machines. Each type follows strictly the latest machine configurations of Amazon EC2 instances, as listed in Table 7. Here, the "vCPU" parameter indicates the processing speed $ms$ of each machine. For example, task $O_{ij}$ with task workload as $tw_{ij} = 402.3$ will require an execution time of $et_{ij}^q = 402.3/8$ on machine $M_q$ equipped with 8 vCPUs. The number of machines for each configuration is determined by the specific scenario. For example, a scenario denoted as $\langle 4 \times 3, 9 \rangle$ comprises of the first four VM types in Table 7, with three machines of each type, resulting in a total of 12 machines.

---

[1]https://aws.amazon.com/ec2/pricing/on-demand/

[2]https://cloud.google.com/compute/vm-instance-pricing

[3]https://azure.microsoft.com/en-gb/pricing/

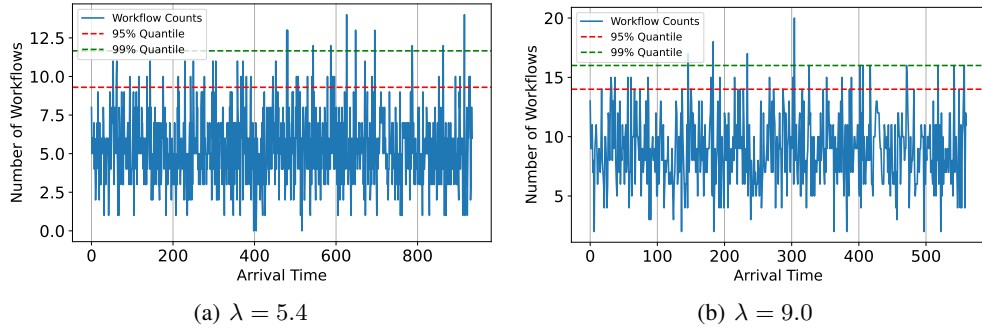

(a) $\lambda = 5.4$                                (b) $\lambda = 9.0$

Figure 10: **New arrival counts of 95%-99% Quantiles at different arrival rates.**

Table 7: **Configurations of six machines based on Amazon EC2.** "vCPU": used as the processing speed of the machine.

| Instance Name | vCPU | Memory | On-Demand hourly rate |
|---|---|---|---|
| m5.2xlarge | 8 | 32 GiB | $0.344 |
| m5.4xlarge | 16 | 64 GiB | $0.688 |
| m5.8xlarge | 32 | 128 GiB | $1.376 |
| m5.12xlarge | 48 | 192 GiB | $2.064 |
| m5a.16xlarge | 64 | 256 GiB | $2.752 |
| m5a.24xlarge | 96 | 384 GiB | $4.128 |

## J  MODEL CONFIGURATIONS

**Network Architecture.** The actor network $\pi$ consists of 2 Graph Attention Network (GAT) layers with one attention head, followed by 4 MLP layers. The critic network $V$ has 2 GAT layers with one attention head, 1 self-attention layer with two attention heads, and 4 MLP layers. Each of these layers has 128 hidden dimensions.

**Normalization.** All raw feature values are normalized by dividing by constants to maintain a consistent scale across the network inputs. The reward $r_t$ is normalized by dividing it by 1000 to ensure stability during training.

**Imitation Learning.** The actor network is pre-trained with 7680 state-transition samples obtained by using the HEFT heuristic. The mini-batch size is set to 64. The actor is updated for 10 epochs using the Adam optimizer with a constant learning rate of $1 \times 10^{-4}$. The choice of 10 epochs for pre-training the actor network was based on experimental observations. After 10 epochs, the actor was able to match or even surpass the performance of the HEFT heuristic.

Table 8: **Imitation learning hyperparameters.**

| Hyperparameter | Value |
|---|---|
| Data size ($|\mathcal{D}|$) | 7680 |
| Num. epochs | 10 |
| Minibatch size | 64 |
| Actor learning rate | $1 \times 10^{-4}$ |

**Offline PPO.** We warm up the critic network for 200 iterations before running 1000 training iterations. Each iteration collects 4 independent episodes of transition data for actor and critic training. Following the implementation in (Barhate, 2021), we use a clipping parameter of 0.2 for PPO. The coefficients for the clipped policy loss and the value function loss are set to 1 and 0.5, respectively. The discount factor $\gamma$ is 0.99. We use the Adam optimizer with fixed learning rates: $3 \times 10^{-4}$ for the actor and $1 \times 10^{-3}$ for the critic. In each iteration, both the actor and critic networks are updated for 1 epoch.

**Online PPO.** For online learning, we update the pre-trained actor network over a single long trajectory with approximately $30\times$ 5000, $30\times 10000$, or $30\times 20000$ steps. After a system warm-up phase of 15000 or 30000 steps, we begin running the online PPO algorithm. We first train the critic network for 50 iterations. Subsequently, both the actor and critic networks are trained for additional 250, 500, or 1000 training iterations. In each iteration, 512 steps of transition data are collected to

Table 9: **Offline PPO hyperparameters.**

| Hyperparameter | Value |
|---|---|
| Critic warm-up iterations | 200 |
| Max training iterations | 1000 |
| Num. independent episodes | 4 |
| Num. epochs | 1 |
| Minibatch size | 64 |
| Clipping parameter | 0.2 |
| Policy coefficient | 1 |
| Value function coefficient | 0.5 |
| Discount factor ($\gamma$) | 0.99 |
| Actor learning rate | $3 \times 10^{-4}$ |
| Critic learning rate | $1 \times 10^{-3}$ |

alternately train the actor and critic networks, with a mini-batch size of 64. The actor and critic are updated for 5 and 20 epochs per iteration, respectively. We use the Adam optimizer with learning rates of $5 \times 10^{-5}$ for the actor and $1 \times 10^{-4}$ for the critic. A gradient control mechanism ensures that the gradient magnitude for each mini-batch stays within the range of the mean plus one standard deviation, based on the previous iteration's values, and does not exceed 0.075.

Table 10: **Online PPO hyperparameters.**

| Hyperparameter | Value |
|---|---|
| Critic warm-up iterations | 50 |
| Max training iterations | 250, 500, 1000 |
| Windows stepsize | 512 |
| Num. epochs for actor | 5 |
| Num. epochs for critic | 20 |
| Minibatch size | 64 |
| Clipping parameter | 0.2 |
| Discount factor ($\gamma$) | 0.99 |
| Actor learning rate | $5 \times 10^{-5}$ |
| Critic learning rate | $1 \times 10^{-4}$ |
| Maximum available gradient change ($\tau_0$) | 0.075 |

**Hardware platform and Algorithm Implementation.** We implement the GAT networks using PyTorch-Geometric (Fey & Lenssen, 2019), and the rest of the components are built using PyTorch (Paszke et al., 2019). Our experiments are conducted in a cloud computing environment to parallel the execution of numerous experiments. Due to the predominance of CPU resources in the cloud environment, all experiments were conducted on CPU. The averaged offline training time was approximately 150 CPU hours, while the averaged online training times were 30 CPU hours, 60 CPU hours, and 120 CPU hours, respectively. Our code and simulator will be made publicly available.

Table 11: **Software and hardware version information.**

| Software and hardware | Version |
|---|---|
| Python | 3.11.5 |
| PyTorch | 2.4.1 |
| PyTorch-Geometric | 2.5.3 |
| rl-zoo3 | 2.3.0 |
| deap | 1.4.1 |
| Nodes | 56 |
| CPUs/Node | 256 |
| Available Mem/CPU | 1850 MB |

## K    OFFLINE PERFORMANCE OF GENETIC PROGRAMMING

Table 12 presents the mean and standard deviation results of the top three GPHH heuristics across various offline scenarios. For each scenario, the corresponding result is averaged over 30 problem

instances with each having 1000, 3000 or 5000 workflows. We select the best results in each scenario (summarized in the right column) and present them in Table 1. As evidenced in this table, the performance of GPHH varies significantly across different scenarios, indicating that the scheduling heuristic evolved by GPHH cannot reliably handle large-scale problems. Previous research on GPHH for DWS has typically focused on small-scale problems (Xu et al., 2023), often with around 30 workflows, where GPHH performed well. However, these small-scale problems are not popular in many cloud computing applications (Huang et al., 2022; Zhu et al., 2024). Our results suggest that GPHH, when applied to large and more complex scenarios, can suffer from significant instability.

Table 12: **Mean and standard deviation objective of GPHH in offline scenarios.** The mean performance of the top three GPHH heuristics on the validation set, evaluated across 30 instances.

| Scenarios | GPHH-top 1 | GPHH-top 2 | GPHH-top 3 | minimum |
|---|---|---|---|---|
| $\langle 5 \times 5, 5.4, 1k \rangle$ | 435.56 | 607.30 | **408.24** | 408.24 |
| $\langle 5 \times 5, 9, 1k \rangle$ | **430.28** | 267913.55 | 52435.90 | 430.28 |
| $\langle 6 \times 4, 5.4, 1k \rangle$ | 726.23 | 380.29 | **322.52** | 322.52 |
| $\langle 6 \times 4, 9, 1k \rangle$ | 5314.90 | **300.20** | 352.45 | 300.20 |
| $\langle 5 \times 5, 5.4, 3k \rangle$ | 435.27 | 622.15 | **407.81** | 407.81 |
| $\langle 5 \times 5, 9, 3k \rangle$ | **427.04** | 1503503.96 | 21490.23 | 427.04 |
| $\langle 6 \times 4, 5.4, 3k \rangle$ | 759.19 | **386.77** | 484.37 | 386.77 |
| $\langle 6 \times 4, 9, 3k \rangle$ | 7277.48 | 7277.48 | **358.40** | 358.40 |
| $\langle 5 \times 5, 5.4, 5k \rangle$ | 437.32 | 622.01 | **408.38** | 408.38 |
| $\langle 5 \times 5, 9, 5k \rangle$ | **427.88** | 2650402.61 | 13750.89 | 427.88 |
| $\langle 6 \times 4, 5.4, 5k \rangle$ | 728.35 | **386.95** | 590.84 | 386.95 |
| $\langle 6 \times 4, 9, 5k \rangle$ | 7504.89 | **297.40** | 360.58 | 297.40 |

## L   ABLATION STUDY OF ONLINE LEARNING METHOD

In Table 13, we validate two key techniques proposed for online training: **gradient control** and **independent high-frequency critic updates**, as described in Subsection 4.3.2. We compare **Ours-Online** against two baselines: **Online w/o grad.** (removing gradient control) and **Online w/o freq.** (removing high-frequency critic updates).

- **Ours-Online** achieves consistent improvements in mean flowtime, demonstrating the stability and effectiveness of both techniques in enhancing online learning performance.
- **Online w/o grad.** shows slight deterioration, highlighting the importance of gradient control for stabilizing actor updates during online training.
- **Online w/o freq.** performs significantly worse, with drastic negative impacts, indicating that frequent critic updates are crucial for providing accurate value estimates and guiding effective policy updates.

Overall, the inclusion of both techniques in Ours-Online ensures stable and improved performance in online learning.

Table 13: **Improvement in mean flowtime compared to Ours-Offline at different iterations.** "Ours-Online": our online learning method proposed in Section 4.3.2. "Online w/o. grad.": our online learning method removes the gradient control technique. " Online w/o. freq.": actor and critic are updated together using an aggregation loss.

| Training Method | 150-th | 175-th | 200-th | 225-th | 250-th |
|---|---|---|---|---|---|
| Ours-Online | **1.62%** | **1.50%** | **1.57%** | **1.52%** | **1.52%** |
| Online w/o. grad. | -1.18% | -1.08% | -1.24% | -1.36% | -1.64% |
| Online w/o. freq. | -184.80% | -261.27% | -283.93% | -336.86% | -382.54% |

## M   SCALABILITY TO SIGNIFICANT CHANGES

We conduct additional experiments to test the generalization ability of our scheduling policy (i.e., actor network) without retraining, focusing on scenario changes in workflow patterns, workflow

arrival rates, and cloud configurations. The results demonstrate that our model can effectively handle variations in workflow patterns, arrival rates, and machine combinations.

As shown in Table 14, we evaluated scenarios with significant differences from the training environment. Here, the "–" symbol indicates that the setting remains the same as the original scenario, while "$\sqrt{}$" denotes a change. The original training scenario involved mixed workflow patterns, $\lambda = 9.0$, and a set of $5 \times 5$ machines. Scenarios 1 and 2 involve only compute-intensive workflow patterns (20 times larger in workload than normal ones), showing that our model maintains strong performance under significant changes in workflow patterns. Scenarios 3 to 6 involve variations in the combination of machines (i.e., configurations $\times$ each quantity), demonstrating our model's ability to adapt to changes in cloud configurations.

Table 14: **Varied scenarios in workflow patterns, arrival rates, and machine numbers.**

| Scenarios | Workflow Pattern | Arrival Rate | Machine Number | EST | PEFT | HEFT | GP | ERL-DWS | Ours |
|---|---|---|---|---|---|---|---|---|---|
| 1 | $\sqrt{}$ | – | – | 1954.59 | 961.26 | 881.55 | 962.35 | 14103.84 | **862.59** |
| 2 | $\sqrt{}$ | $\sqrt{}$ | – | 2114.21 | 1005.76 | 904.06 | 832.37 | 6403.65 | **791.86** |
| 3 | – | $\sqrt{}$ | $3 \times 15$ | 1793.76 | 927.33 | 872.71 | 1015.96 | 3208.32 | **761.24** |
| 4 | – | $\sqrt{}$ | $4 \times 10$ | 1512.44 | 684.15 | 643.34 | 517.05 | 2696.69 | **509.17** |
| 5 | – | $\sqrt{}$ | $5 \times 7$ | 1317.28 | 561.51 | 513.70 | 396.07 | 2534.30 | **385.44** |
| 6 | – | $\sqrt{}$ | $6 \times 5$ | 1190.84 | 450.93 | 404.47 | 286.00 | 2420.63 | **282.07** |

## N    TRANSFERABILITY TO OTHER SCHEDULING PROBLEMS

To demonstrate the transferability of our method, we applied our graph representation and neural network architecture to the FJSS problems studied in Song et al. (2022), chosen for its strong representativeness. In our experiments, all sequence-structured jobs of an FJSS problem instance are represented jointly as a workflow, enabling our pipeline to handle the FJSS problem effectively as a special case of our workflow scheduling problem.

As summarized in Table 15, GOODRL achieved highly competitive performance compared to DRL-G and DRL-S proposed in Song et al. (2022) on the same FJSS test instances with different sizes. Results confirms that GOODRL can be transferred to solve related scheduling problems, including FJSS problems. It is important to note that GOODRL is purposefully designed to address dynamic scheduling problems, leveraging key innovations such as task-specific and system-oriented graph representations, which are tailored to handle unpredictable workflow arrivals and evolving system states. Existing research such as (Zhang et al., 2020; Song et al., 2022; Zhang et al., 2024; Huang et al., 2024) mainly considered static problems.

Table 15: **Results on FJSS instance with different sizes.** "DRL-G" and "DRL-G" are methods proposed in (Song et al., 2022).

| FJSS Size | MOR | SPT | FIFO | MWKR | DRL-G | DRL-S | Ours |
|---|---|---|---|---|---|---|---|
| $10 \times 5$ | 116.69 | 129.06 | 119.62 | 115.29 | 111.67 | **105.61** | 112.57 |
| $20 \times 5$ | 217.17 | 229.89 | 216.13 | 216.98 | 211.22 | 207.50 | **202.38** |
| $30 \times 10$ | 320.18 | 347.40 | 328.50 | 319.89 | 313.04 | 312.20 | **304.63** |
| $40 \times 10$ | 425.19 | 443.30 | 427.22 | 425.70 | 416.18 | 415.15 | **395.70** |

## O    EXPANDABILITY TO MULTI-OBJECTIVE PROBLEMS

We conduct experiments to demonstrate our method can support other practical objectives beyond flowtime reduction, typically by modifying the reward function. By directly incorporating a cost term into the reward function, we can re-train the actor to optimize for both VM cost (calculated based on the hourly rental fees of each VM type) and flowtime. The results in the table below indicate that, when explicitly considering the cost of using VMs, the mean flowtime experiences a slight increase of up to 8%, but substantial cost savings of up to 41% are achieved in some scenarios.

Hence, with a modified reward function, GOODRL can achieve a desirable trade-off between flow-time and cost, showcasing its flexibility and practical utility. These results differ from our original focus on reducing the mean flowtime alone and demonstrate the potential of GOODRL to support a wide range of objectives.

Table 16: **Performance comparison of policies trained with single and multi-objective.**

| Scenarios | Objectives | Single-Obj. | Multi-Obj. | Diff. |
|---|---|---|---|---|
| $\langle 5 \times 5, 5.4, 30 \rangle$ | *flowtime* | 401.77 | 420.29 | +4.61% |
| | *cost* | 139.82 | 82.28 | -41.15% |
| $\langle 5 \times 5, 5.9, 30 \rangle$ | *flowtime* | 408.49 | 413.02 | +1.11% |
| | *cost* | 116.32 | 97.51 | -16.17% |
| $\langle 6 \times 4, 5.4, 30 \rangle$ | *flowtime* | 277.57 | 286.73 | +3.30% |
| | *cost* | 192.24 | 143.47 | -25.37% |
| $\langle 6 \times 4, 9, 30 \rangle$ | *flowtime* | 285.93 | 306.90 | +7.33% |
| | *cost* | 135.58 | 91.18 | -32.75% |

Extending our approach to incorporate multi-objective optimization is both feasible and valuable. Below, we outline how this can be achieved in general and the specific challenges that need to be addressed:

- **Reward design:** A weighted reward combining flowtime and energy efficiency (or cost as a proxy) can be used. Weights could be static or adaptively tuned.
- **Graph representation:** Additional features like machine prices, scheduling overhead, and other Quality of Service (QoS) requirements can be incorporated to provide the necessary context for optimizing both objectives.
- **Learning challenges:** Conflicting objectives require strategies like Pareto-optimal training or multi-critic designs to balance trade-offs effectively.

## P    INFERENCE TIME COMPARISON

At inference time, our model (i.e., the task-specific actor network) requires a very short time to make a decision, as reported in Table 17. Across various test scenarios, our model takes 6-7 *ms* to allocate a VM to process the target task. While prior approaches such as ERL-DWS and GPHH are faster, it is important to emphasize that our model's inference time is sufficiently short to meet the requirements of real-world DWS problems. In fact, the inference time is significantly less than the typical communication latency and data transfer time in a cloud environment.

Table 17: **The average inference time to make a decision.**

| Scenarios | GPHH | ERL-DWS | Ours |
|---|---|---|---|
| $\langle 5 \times 5, 5.4, 1k \rangle$ | 0.7 ms | 2.6 ms | 6.1 ms |
| $\langle 5 \times 5, 9, 1k \rangle$ | 1.0 ms | 2.7 ms | 7.6 ms |
| $\langle 6 \times 4, 5.4, 1k \rangle$ | 0.6 ms | 2.7 ms | 6.0 ms |
| $\langle 6 \times 4, 9, 1k \rangle$ | 0.7 ms | 2.5 ms | 6.8 ms |

## Q    ROBUSTNESS ANALYSIS IN ONLINE SCENARIOS

(1) **Fluctuation of Workflow Arrivals**: We conducted additional experiments where we introduced significant changes to the arrival rate, ranging from a decrease of 50% to an increase of 100% relative to the arrival rates used to train the actor network. As evidenced by the experiment results reported in Table 18, GOODRL can achieve consistently the best performance, outperforming all competing approaches by a large margin. These results demonstrate that GOODRL is robust to substantial fluctuations in arrival patterns and can effectively adapt to dynamic changes over time.

(2) **Fluctuation of Offline-trained Actor**: We have conducted additional experiments to investigate whether our online algorithm can quickly adapt to such performance deterioration. In our experiments, we introduced artificial noise $\epsilon$ at various levels to the offline-trained actor to intentionally

Table 18: **Performance comparison under changed arrival rates.**

| Arrival Rates | EST | PEFT | HEFT | GP | ERL-DWS | Ours |
|---|---|---|---|---|---|---|
| -50% | 1288.59 | 626.37 | 567.55 | 403.99 | 2328.24 | **398.01** |
| +50% | 1165.15 | 516.60 | 481.70 | 413.99 | 4076.43 | **409.66** |
| +100% | 1112.05 | 498.06 | 469.46 | 424.92 | 5255.60 | **423.02** |

degrade its performance. We then trained this noise-infused actor online over a series of iterations. As shown in Table 19, the initial performance dropped by approximately 3.7–3.8% compared to the actor without noise. However, after 150 online training iterations, this gap was significantly reduced to around 1–2%. These results clearly demonstrate that our online learning method in GOODRL can quickly adapt to distribution shifts and effectively recover performance, even when starting with a suboptimal offline-trained actor.

Table 19: **Performance gap with the no-nosie actor across online training iterations.**

| Noise | 0-th | 25-th | 50-th | 75-th | 100-th | 125-th | 150-th |
|---|---|---|---|---|---|---|---|
| $\epsilon=0.05$ | 3.71% | 3.61% | 3.49% | 3.20% | 2.21% | 1.94% | 1.95% |
| $\epsilon=0.1$ | 3.87% | 3.66% | 2.92% | 2.42% | 1.46% | 1.10% | 1.06% |

# R   ADVANTAGES OF ONLINE RL COMPONENT

The adoption of the Online RL component is important for ensuring long-term stability and practical utility in large-scale DWS problems:

- **Maintain high long-term utility:** The online RL component enhances long-term utility of GOODRL by constantly learning from new experiences collected from the daily operation of the cloud system. Unlike offline learning, online RL avoids costly and unnecessary retraining of the actor from scratch. It is highly sample efficient and responsive, offering strong adaptability in practice.

- **Practical significance:** The small improvement in mean flowtime can translate to a significant improvement in scheduling efficiency and resource utilization for large-scale problems. As shown in Table 20, even small improvements in flowtime (e.g., $0.84\%$ reduction in average flowtime) can result in substantial cost savings (e.g., $36.11\%$ reduction in machine rental fees). We will include quantitative examples in the revised paper to clearly highlight the practical significance of this improvement.

Table 20: **Comparison of two scheduling plans in flowtime and cost.**

| Scenarios | Objectives | Scheduling Plan-1 | Scheduling Plan-2 | Diff. |
|---|---|---|---|---|
| $\langle 5 \times 5, 5.4, 30 \rangle$ | *flowtime* | 421.94 | 420.29 | 0.39% ↓ |
|  | *cost* | 102.44 | 82.28 | 19.68% ↓ |
| $\langle 5 \times 5, 9, 30 \rangle$ | *flowtime* | 416.53 | 413.02 | 0.84% ↓ |
|  | *cost* | 152.63 | 97.51 | 36.11% ↓ |
| $\langle 6 \times 4, 5.4, 30 \rangle$ | *flowtime* | 292.81 | 286.73 | 2.08% ↓ |
|  | *cost* | 188.43 | 143.47 | 23.86% ↓ |
| $\langle 6 \times 4, 9, 30 \rangle$ | *flowtime* | 309.70 | 306.90 | 0.90% ↓ |
|  | *cost* | 137.43 | 91.18 | 33.65% ↓ |

# S   RELATED WORDS ON LEARN-TO-OPTIMIZE

The Learn-to-Optimize (L2O) field extensively uses Graph Neural Networks (GNNs) and Reinforcement Learning (RL) to solve combinatorial optimization problems, including Vehicle Routing Problems (VRP) (Bi et al., 2024; Hou et al., 2023; Zhou et al., 2024), Job Shop Scheduling (JSS) (Zhang et al., 2020; 2024; Park et al., 2021; Su et al., 2023; Huang et al., 2024), and Multi-Agent Task Allocation (MATA) (Altundas et al., 2022; Wang & Gombolay, 2020). However, the specific

graph representations and network architectures used in L2O approaches must be tailored to the unique characteristics and challenges of each problem domain.

*These domain-specific adaptations often constitute the core contributions of L2O research.* In this context, our research introduces novel graph representations and network architectures specifically designed to tackle the unique challenges of dynamic workflow scheduling (DWS) problems, clearly differentiating it from existing L2O approaches. For example, while encoder-decoder models are commonly used for VRPs, scheduling problems must handle complex operation-job-machine constraints, making such models less effective, as noted in Zhang et al. (2020).

In L2O research for scheduling:

- Studies in (Zhang et al., 2020; Park et al., 2021; Zhang et al., 2024) tackled static JSS, where Zhang et al. (2020) used a topological graph representation directly trained with PPO, and Zhang et al. (2024) enhanced this with an additional context-aware embedding module to improve the actor's ability to distinguish actions. Park et al. (2021) incorporated raw features for static JSS to improve representation.

- The work you mentioned in Song et al. (2022) addressed flexible JSS (FJSS), which introduced a heterogeneous graph representation and the corresponding network architecture design, directly trained with PPO.

- For dynamic JSS, Su et al. (2023) proposed new graph representations by considering stochastic processing times in raw features, and machine failures in action selection, and directly used the ESRL method. However, it still focuses on small-scale problems, with unchangeable graph, i.e., the number of nodes on the graph is fixed over time.

- The work in Huang et al. (2024) focused on distributed JSS, which also used PPO and claimed their contributions on graph representations for solving challenges in distributed JSS but with small-scale.

Thus, while these methods all leverage GNNs and RL, addressing the challenges in different problem variants still requires unique designs and careful considerations, making each approach valuable to the research community. As Zhang et al. (2024) highlights, "Unlike routing problems that are vastly studied, JSS has received relatively less attention, and the performance of existing learning-based solvers is still far from optimal due to the lack of an effective learning framework and neural representation scheme".

Our work focuses on **Dynamic Workflow Scheduling** (DWS), which not only presents a more complex setting but also better aligns with real-world applications. It has great potential to push the boundary of using DRL to solve real-world COPs, especially scheduling problems, in several key aspects:

- **Dynamic and evolving states:** Workflow arrivals and patterns are unpredictable. Consequently, the number of active workflows varies significantly across time, unlike the fixed setup in previously studied scheduling problems. This fluctuation in active workflows introduces unpredictable state transitions, requiring new methods to effectively handle their impact on learning stability and performance.

- **Large problem size:** Our approach can handle problem instances involving up to 600,000 workflow tasks, which is more than 100 times larger than previously studied JSS problems (Zhang et al., 2020; 2024; Park et al., 2021; Su et al., 2023; Huang et al., 2024) and MATA problems (Altundas et al., 2022; Wang & Gombolay, 2020). Furthermore, Cloud service providers require scheduling policies that can continuously adapt to new operating conditions, necessitating online updates to policy network parameters.

- **Workflow-specific constraints and heterogeneous machines:** Unlike JSS, DWS involves intricate inter-task dependencies, flexible machine choices, and highly heterogeneous processing time. These challenges demand for a novel design of the graph representations that can properly capture intricate relationship among tasks, workflows, and machines.

In view of the above challenges, existing graph representations and network designs for small-scale or static JSS and MATA problems are insufficient for DWS.

A recent survey (Jayanetti et al., 2024) from a world-leading cloud computing research group revealed that existing RL-based methods for workflow scheduling often relied on vector or matrix representations, lacking effective GNN-based methods for this domain. Additionally, solving large-scale dynamic problems with PPO requires novel algorithmic designs to significantly enhance learning stability, which is often overlooked in prior studies.

Building on the above discussion, our paper makes significant contributions to the existing literature on scheduling problems in:

- Designing problem-specific graph representations and network architectures to tackle the unique challenges of large-scale DWS problems, which have been largely overlooked in prior research; and
- Proposing a novel offline-online learning framework to ensure high stability and adaptability in dynamic DWS environments.

