# OpenReview forum: "Graph Assisted Offline-Online Deep Reinforcement Learning for Dynamic Workflow Scheduling"
_ICLR.cc/2025/Conference — ICLR 2025 Poster_

### Official Review · Reviewer_RdYS · 2024-11-01

**Soundness:** 3
**Presentation:** 3
**Contribution:** 3
**Rating:** 6
**Confidence:** 3

**Summary:**

The paper introduces an innovative approach, GOODRL, for handling dynamic workflow scheduling in cloud computing environments. This method integrates a task-specific graph representation with Graph Attention Networks (GATs) for actor-critic networks and incorporates offline imitation learning alongside online reinforcement learning to adapt to changing conditions. The proposed system is evaluated in various online and offline scenarios and is shown to outperform existing state-of-the-art methods in terms of mean flowtime and adaptability.

**Strengths:**

- The paper proposes a unique combination of graph representations tailored for actor and critic networks, enhancing action differentiation and value estimation.
- GOODRL demonstrates improvements in mean flowtime over established baseline algorithms, showcasing robust performance in both offline and online settings.
- The offline-online learning approach with imitation learning and gradient control addresses challenges in adapting to dynamic environments, adding practical value.
- The ablation studies and performance comparisons are thorough, providing strong evidence for the contributions and architectural decisions.

**Weaknesses:**

- The paper's reliance on simulations limits its generalizability to real-world cloud environments. Practical tests in real cloud data centers would bolster the validity of the results.
- The experiments primarily focus on a specific set of workflow types and machine configurations, potentially limiting the applicability of findings to other types of DWS problems.
- The computational overhead associated with the proposed GAT-based architectures is not discussed in detail, raising questions about deployment feasibility in large-scale, real-time applications.
- While the method performs well in flowtime reduction, other practical objectives, such as energy efficiency and cost, are not explored, which would be valuable for broader applicability.
- The paper lacks discussion on how the model generalizes to varied workloads, impacting its robustness in dynamic cloud environments.

**Questions:**

- Could the authors elaborate on how their model adapts to significant changes in workflow patterns or cloud configurations without extensive retraining? How does the method maintain robust performance under dynamic conditions that differ from the training data?
- What are the practical deployment requirements for GOODRL in terms of computational resources, and how do they compare with simpler heuristic-based solutions in large-scale, real-time applications?
- Can the authors provide more insights into how the method handles noisy or incomplete data, which is a common challenge in real-world cloud scheduling environments?
- How might the proposed approach be extended or adapted to incorporate multi-objective optimization, such as balancing energy efficiency with flowtime reduction, and what specific challenges would need to be addressed to achieve this?
- Could the authors comment on potential scalability issues when deploying GOODRL in larger cloud infrastructures or in environments with highly heterogeneous machine configurations, and how these challenges could be mitigated?

---

> ### Author Response · Authors · 2024-11-20
> **Response to Reviewer RdYS**
>
> We thank the reviewer for recognizing the significance of our work. We value your feedback and hope to address your concerns by following responses.
>
> **W1: Practical tests in real data centers would bolster the validity of the results.**
>
> We agree that testing in real data centers would offer valuable validation of our approach. However, such experiments are **highly costly** and require **substantial investment in human labor**, making them **impractical** for us at this stage. This limitation applies to **most related studies** . We appreciate your suggestion and will consider pursuing real-world testing in **future research if sufficient funding and resources become available**.
>
> _Simulations remain the standard and mainstream methodology in this field._ For example, the **CloudSim simulator** [1] introduced by a world-leading cloud computing research group has **been cited 6,449 times** to date. We believe simulations do not significantly limit our approach's generalizability, as they are **carefully designed to mimic real-world cloud dynamics** like unpredictable workflow arrivals, resource heterogeneity, and dynamic operating conditions. Benchmarking against SOTA methods demonstrates our approach's robustness, indicating it would perform well in real-world environments.
>
> **W2: Experiments on specific workflows and configurations may limit applicability to other DWS problems.**
>
> Regarding specific workflows, we considered all the popularly studied scientific workflows [2] in our experiments, ensuring a fair comparison with baseline methods. We believe that _focusing on these workflows does not limit the applicability of our findings_ but instead **ensures a robust and meaningful evaluation** of our approach within a common experimental framework.
>
> Regarding machine configurations, we used **real-world machine configurations** from Amazon EC2, the largest global cloud provider, reflecting practical settings in cloud environments. While experiments with other providers could enhance generalizability, we believe _using EC2 configurations does not limit the applicability_. We appreciate the reviewer’s suggestion and will consider incorporating such experiments in future research.
>
> Notably, our experiments address **much larger** problem instances (up to 600,000 decision points) than many existing studies. Moreover, considered dynamic conditions **align with real-world cloud** scenarios. This scale highlights the broad applicability of our approach, and our strong performance under these challenging conditions supports its relevance to dynamic workflow scheduling research.
>
> **W3: The computational overhead associated with the proposed method.**
>
> We will update the manuscript to include a discussion of the computational costs of our method. In Table R1, we compared the decision-making time of three types of architectures on a single CPU core. While our GAT-based approach has a higher decision time than GPHH and ESRL, it remains within a reasonable **6-7 millisecond** range, **comparable to cloud data transmission latency, making it suitable for real-time applications**.
> | Scenarios                    | GPHH   | ESRL   | Ours   |
> |------------------------------|--------|--------|--------|
> | ⟨5×5,5.4,1k⟩ | 0.7 ms | 2.6 ms | 6.1 ms |
> | ⟨5×5,9,1k⟩  | 1.0 ms | 2.7 ms | 7.6 ms |
> | ⟨6×4,5.4,1k⟩  | 0.6 ms | 2.7 ms | 6 ms   |
> | ⟨6×4,9,1k⟩  | 0.7 ms | 2.5 ms | 6.8 ms |
>
> **Table R1. The computational overhead to make a decision.**
>
> ---
> [1] Calheiros, R. N., Ranjan, R., Beloglazov, A., De Rose, C. A., \& Buyya, R. (2011). CloudSim: a toolkit for modeling and simulation of cloud computing environments and evaluation of resource provisioning algorithms. _Software: Practice and experience.
>
> [2] Deelman, E., Vahi, K., Juve, G., Rynge, M., Callaghan, S., Maechling, P.J., Mayani, R., Chen, W., Da Silva, R.F., Livny, M. \& Wenger, K. (2015). Pegasus, a workflow management system for science automation. _Future Generation Computer Systems_.

---

> ### Author Response · Authors · 2024-11-20
> **Response to Reviewer RdYS cnt.**
>
> **W4: Exploring other practical objectives to enhance the method's broader applicability.**
>
> We confirm that our method **can support other practical objectives** beyond flowtime reduction, typically by modifying the reward function.
>
> Table R2 gives examples of considering cost as another objective. We can re-train the actor with a modified reward function to optimize for both **cost** (i.e., machine rental fees) and **flowtime**. Results show that incorporating costs leads to a _slight increase_ in mean flowtime (up to 8%) but achieves **significant cost savings** of up to 41% in some scenarios. Thus, with a modified reward function, GOODRL can **achieve a desirable trade-off** between flowtime and cost, highlighting its broader applicability.
> | Scenarios | Objectives  | Single-Obj. | Multi-Obj. | Diff.       |
> |-----------|-------------|-------------|------------|-------------|
> | ⟨5×5,5.4,30⟩ | *flowtime* | 401.77      | 420.29     | +4.61%     |
> |     | *cost*     | 139.82      | 82.28      | -41.15%    |
> | ⟨5×5,9,30⟩ | *flowtime* | 408.49      | 413.02     | +1.11%     |
> |     | *cost*     | 116.32      | 97.51      | -16.17%    |
> | ⟨6×4,5.4,30⟩ | *flowtime* | 277.57      | 286.73     | +3.30%     |
> |      | *cost*     | 192.24      | 143.47     | -25.37%    |
> | ⟨6×4,9,30⟩ | *flowtime* | 285.93      | 306.90     | +7.33%     |
> |      | *cost*     | 135.58      | 91.18      | -32.75%    |
>
> **Table R2. Performance comparison of policies trained with single and multi-objective.**
>
> We recognize the importance of considering **energy efficiency** in cloud scheduling. However, since _energy consumption depends on the workload of physical machines (PMs)_ managed by cloud providers, it falls outside the scope of our workflow scheduler.
>
> If future opportunities allow us to collaborate with cloud providers or access VM-to-PM allocation data, we will consider integrating energy efficiency into our study to further broaden the applicability of GOODRL. The above discussions and experiments will be updated in our paper.
>
> **W5: Discussion on how the model generalizes to different workloads.**
>
> We agree that discussing the model's generalization to varied workloads would enhance the paper and highlight its robustness.
>
> Table 2 in our paper shows that the pre-trained **GOODRL policy** performs well across scenarios with **varied workloads** or **machine numbers**, consistently outperforming baselines and **demonstrating strong generalization**.
>
> Our model achieves robustness in dynamic cloud environments in two key aspects:
> - **Dynamic Graph Representation**: The proposed dynamic graph structure **continuously updates** based on real-time system states (e.g., varying number of workflows or machine load conditions), enabling it to effectively capture varied workloads.
> - **Offline-Online Training Method**：This method enables the offline-trained actor to **continuously enhance its performance** by learning from online experiences during daily operations in dynamic cloud environments.
>
> We will enhance this discussion with experiments demonstrating the model's adaptability to varied workloads. Detailed results are included in **the response of Q1** and will be reflected in the updated paper.
>
> **Q1: Model adaption to significant changes in workflow patterns or cloud configurations without extensive retraining.**
>
> We conducted additional experiments to evaluate the actor network's generalization to changes in workflow patterns, arrival rates, and cloud configurations without retraining.
>
> | Scenarios | Wf. | Arr. | Mach. | EST     | PEFT    | HEFT    | GP      | ESRL    | Ours    |
> |-----------|----|-----|------|---------|---------|---------|---------|---------|---------|
> | 1  | ✓  | --  | --   | 1954.59 | 961.26  | 881.55  | 962.35  | 14103.8 | **862.6** |
> | 2 | ✓  | ✓   | --   | 2114.21 | 1005.76 | 904.06  | 832.37  | 6403.65 | **791.9** |
> | 3  | -- | ✓   | 3×15 | 1793.76 | 927.33  | 872.71  | 1015.96 | 3208.32 | **761.2** |
> | 4 | -- | ✓   | 4×10 | 1512.44 | 684.15  | 643.34  | 517.05  | 2696.69 | **509.2** |
> | 5  | -- | ✓   | 5×7  | 1317.28 | 561.51  | 513.70  | 396.07  | 2534.30 | **385.4** |
> | 6  | -- | ✓   | 6×5  | 1190.84 | 450.93  | 404.47  | 286.00  | 2420.63 | **282.1** |
>
> **Table R3. Varied scenarios in workflow patterns, arrival rates, and machine numbers.**
>
> Results in Table R3 show it effectively handles these variations.
> - *Scenarios 1 and 2*, with **only compute-intensive workflow patterns** (i.e., 20 times larger than normal), show that our model performs well under significant workflow changes.
> - *Scenarios 3 to 6*, with **variations in machine configurations** (i.e., configurations × each quantity), demonstrate the model's adaptability to cloud configuration changes.
>
> Our model maintains robust performance under diverse conditions due to its **dynamic graph representation** and **offline-online training** method. We will include these experiments and discussions in the revised paper.

---

> ### Author Response · Authors · 2024-11-20
> **Response to Reviewer RdYS cnt.**
>
> **Q2: Practical deployment requirements for GOODRL in terms of computational resources.**
>
> The core computation of GOODRL is powered by a GNN, which operates efficiently on widely available hardware resources. For example, on a single Intel Xeon CPU core (2.8GHz), it makes scheduling decisions in **6-7 ms**. Such **high computational efficiency** ensures its **suitability for real-time decision-making** in large-scale, dynamic environments.
>
> Heuristic methods like HEFT may reduce decision times to under 1 ms. However, **this minor reduction in _millisecond-level_ has negligible practical impact**, as the time required for executing tasks or communicating between machines dominates the scheduling process.
>
> More importantly, **simple heuristics lack the flexibility** of GOODRL, which can adapt to changing environments and outperform simple heuristics. GOODRL delivers superior scheduling outcomes while remaining computationally feasible, making it highly **practical for real-world deployment**.
>
> **Q3: Insights in handling noisy or incomplete data.**
>
> GOODRL requires only basic workflow information, including inter-task dependencies and computation resource requirements. We interpret “noisy or incomplete data” as cases where newly arrived workflows lack precision or completeness:
>
> - **Inter-task Dependencies**: These must be precise for a workflow to be processed accurately.
>
> - **Computation Resource Requirements**: If noisy, GOODRL may make suboptimal decisions, similar to other methods. Integrating recent **resource prediction** techniques [3-4] may enhance GOODRL's reliability, though this is **beyond the current scope** and will be explored in future work.
>
> - **Incomplete Data**: Missing resource requirements make scheduling tasks unfeasible for any scheduler. To mitigate this, users typically provide **estimated** (often pessimistic) **resource needs**, resulting in over-allocation. Advanced machine learning methods [5-6] can improve these estimates, which GOODRL could adopt to handle such cases better.
>
> We appreciate your suggestion to address this important challenge in our future research.
>
> **Q4: Extend the proposed approach to support multi-objective optimization, such as cost and energy efficiency.**
>
> Extending GOODRL to support multi-objective optimization is feasible and valuable, as explained in response to W4. Key considerations to be covered in the revised paper include:
>
> - **Reward Design**: A weighted reward combining flowtime and energy efficiency (or cost) can be used.
>
> - **Graph Representation**: Additional features, such as machine prices, scheduling overhead, and QoS requirements, can be integrated into graph nodes.
>
> - **Learning Challenges**: Addressing trade-offs between conflicting objectives may require new techniques like Pareto-optimal training [7].
>
> **Q5: Scalability and heterogeneity challenges in large cloud environments.**
>
> GOODRL has been **successfully** tested on **large DWS** problems with up to 600,000 tasks, demonstrating its scalability in this aspect.
>
> While we currently assume a fixed VM collection, expanding this would increase the size of the graph-based state representation. However, prior research showed that GNNs can scale to process graphs with millions of nodes [8-9].
>
> To process large graphs efficiently, we can **restrict the VMs considered for each task** by using heuristic rules or machine learning models to **pre-select suitable VMs for graph construction**. Such approaches will be explored in future work to be discussed in the revised paper. We also note that managing extremely large resource sets is not our current focus and remains an open challenge in the field.
>
> GOODRL is designed to work with **heterogeneous machine configurations** (e.g., VMs provided by Amazon EC2 in our experiments), which are captured by several machine features in the graph representation. Experiments that further evaluate the impact of resource heterogeneity on scheduling performance will be covered in our discussion of future research.
>
> ---
> [3] Ullah et. al. (2023). Intelligent time-series forecasting framework for non-linear dynamic workload and resource prediction in cloud. _Computer Networks_.
>
> [4] Nawrocki et. al.. (2023). Data-driven adaptive prediction of cloud resource usage. _Journal of Grid Computing_.
>
> [5] Dogani et. al.. (2023). Multivariate workload and resource prediction in cloud computing using CNN and GRU by attention mechanism. _The Journal of Supercomputing_.
>
> [6] Jia et. al.. (2024). DuCFF: A dual-channel feature-fusion network for workload prediction in a cloud infrastructure. _Electronics_.
>
> [7] Lin et. al. (2022). Pareto set learning for expensive multi-objective optimization. In _NeurIPS.
>
> [8] Chiang et. al. (2019). Cluster-gcn: An efficient algorithm for training deep and large graph convolutional networks. In _KDD_.
>
> [9] Wu et. al. (2022). Nodeformer: A scalable graph structure learning transformer for node classification. In _NeurIPS_.

---

> > ### Comment · Reviewer_RdYS · 2024-11-26
> >
> > Thank you for the detailed responses, particularly on scalability, multi-objective optimization, and computational overhead. The explanations mostly addressed my concerns. I have revised my score in response.

---

> > > ### Author Response · Authors · 2024-11-26
> > >
> > > We sincerely thank the reviewer for thoroughly reviewing our responses, acknowledging the value of our work, and raising the score!

---

### Official Review · Reviewer_8vKM · 2024-11-04

**Soundness:** 2
**Presentation:** 3
**Contribution:** 1
**Rating:** 8
**Confidence:** 4

**Summary:**

The paper presents Graph assisted Offline-Online Deep Reinforcement Learning (GOODRL) for Dynamic Workflow Scheduling for Cloud Computing. The presence of heterogenous configurations, dynamic arrival of workflows, and constant evolving environment makes this a challenging problem for State of the Art Models.

The Contributions presented in the paper are:

1) A Task-Specific Graph Representation and Graph Attention Actor Model that dynamically assign focused tasks to heterogenous machines.
2) Explicit Consideration of the future impact of the crucial state.
3) A combination of Offline Imitation Learning followed by Online PPO.

**Strengths:**

- Clear and detailed explanation of the approach being used.
- Easy to understand figures.
- Compared against Multiple Benchmarks and show that the model outperforms the baselines in most instances

**Weaknesses:**

- The topic of Resource Opimization using Graph Neural Networks is an open problem that has applications not limited to Cloud Computing. The problem itself is also explored under Multiple Travelling Salesman Problems [1, 2], Vehicle Routing Problem [3], Job Shop Scheduling [4] and Task Allocation and Scheduling in Multi-Agent Systems [5, 6]. While the application of the problem into Cloud Computing is novel, the use of Reinforcement Learning and Graph Attention Networks to similar optimization problems exists.

- It is unclear how the proposed method differs from Online Predictive Scheduling using Heterogenous Graph Attention presented in Wang et al 2022 [7]:

- The enhancement provided by the Online RL part of the model is unclear. The experimental results show that the Offline Learning allows for the model to be within 2% of the Online Training results. The significance of this improvement is unclear and needs to be discussed clearly.

[1] Yujiao Hu, Yuan Yao, and Wee Sun Lee. 2020. A reinforcement learning approach for optimizing multiple traveling salesman problems over graphs. Knowledge-Based Systems 204 (Sept. 2020), 106244. https://doi.org/10.1016/j.knosys.2020. 106244

[2] Yujiao Hu, Zhen Zhang, Yuan Yao, Xingpeng Huyan, Xingshe Zhou, and Wee Sun Lee. 2021. A bidirectional graph neural network for traveling salesman problems on arbitrary symmetric graphs. Engineering Applications of Artificial Intelligence 97 (Jan. 2021), 104061. https://doi.org/10.1016/j.engappai.2020.104061

[3] Steve Paul and Souma Chowdhury. 2022. A scalable graph learning approach to capacitated vehicle routing problem using capsule networks and attention mechanism, Vol. 86236. American Society of Mechanical Engineers, V03BT03A045

[4] Song, Wen, et al. "Flexible job-shop scheduling via graph neural network and deep reinforcement learning." _IEEE Transactions on Industrial Informatics_ 19.2 (2022): 1600-1610.

[5] Z. Wang and M. Gombolay, "Learning Scheduling Policies for Multi-Robot Coordination With Graph Attention Networks," in IEEE Robotics and Automation Letters, vol. 5, no. 3, pp. 4509-4516, July 2020, doi: 10.1109/LRA.2020.3002198.

[6] B. Altundas, Z. Wang, J. Bishop and M. Gombolay, "Learning Coordination Policies over Heterogeneous Graphs for Human-Robot Teams via Recurrent Neural Schedule Propagation," _2022 IEEE/RSJ International Conference on Intelligent Robots and Systems (IROS)_, Kyoto, Japan, 2022, pp. 11679-11686, doi: 10.1109/IROS47612.2022.9981748.

[7] Wang, Z., & Gombolay, M. (2022). Stochastic Resource Optimization over Heterogeneous Graph Neural Networks for Failure-Predictive Maintenance Scheduling. _Proceedings of the International Conference on Automated Planning and Scheduling_, _32_(1), 527-536. https://doi.org/10.1609/icaps.v32i1.19839

**Questions:**

- How does this model differ from the model presented in Wang et al 2022 [7]?

- How does the heterogeneity of the agent-task is accounted for in the graph representation?

- It is unclear what the novelty of this work is compared to similar works published in Multi-Agent Coordination, and Task Allocation and Scheduling Domains.

---

> ### Author Response · Authors · 2024-11-20
> **Response to Reviewer 8vKM**
>
> We thank the reviewer for the detailed feedback. We hope our following response can address your concerns.
>
> **W1: Novelty of applying reinforcement learning and graph attention networks.**
>
> GNNs and RL have been widely applied to combinatorial optimization problems (COPs), including VRP [8-10], JSS [11-15], and MATA [5-6]. However, **each problem domain presents unique challenges that require tailored graph representations and network architectures**, often forming the **core contributions**. For example, VRP studies often use encoder-decoder models [8-10], while JSS and MATA rely on GNNs to handle complex constraints [5-6,11-15].
>
> Our work focuses on Dynamic Workflow Scheduling (DWS), which _has great potential to push the boundary of using DRL to solve real-world COPs, especially scheduling problems_, in several aspects:
>
> - Dynamically **evolving states** represented by graphs with changing structures;
> - **Large problem size** (600,000 tasks), far exceeding the scales of previously studied scheduling problems;
> - **Sophisticated constraints** that arise from intricate inter-task dependencies, flexible machine choices, and highly heterogeneous task processing time.
>
> Existing graph representations and learning frameworks used in JSS or MATA [5-6,11-15] are **insufficient to handle these complexities**. Additionally, a reputable survey [16] highlights **the lack of effective GNN-based methods for workflow scheduling**, with most existing approaches relying on **simpler vector** or **matrix representations.**
>
> Building on these challenges, our contributions include: (1) problem-tailored **graph representations** and **architectures** for **large-scale** DWS, (2) **independently designed** and trained actor and critic; and (3) a **novel offline-online learning** framework ensuring _stability_ and _adaptability_ in dynamic environments. _These innovations push the boundary of using DRL for real-world scheduling problems._ We will clarify these novelties in the revised paper.
>
> ---
> [8] Hou, Q., Yang, J., Su, Y., Wang, X., \& Deng, Y. (2023). Generalize learned heuristics to solve large-scale vehicle routing problems in real-time. In _ICLR_.
>
> [9] Zhou, J., Cao, Z., Wu, Y., Song, W., Ma, Y., Zhang, J., \& Xu, C. (2024). MVMoE: Multi-task vehicle routing solver with mixture-of-experts. In _ICML_.
>
> [10] Bi, J., Ma, Y., Zhou, J., Song, W., Cao, Z., Wu, Y., \& Zhang, J. (2024). Learning to handle complex constraints for vehicle routing problems. In _NeurIPS_.
>
> [11] Zhang, C., Song, W., Cao, Z., Zhang, J., Tan, P. S., \& Chi, X. (2020). Learning to dispatch for job shop scheduling via deep reinforcement learning. In _NeurIPS_.
>
> [12] Zhang, C., Cao, Z., Song, W., Wu, Y., \& Zhang, J. (2024). Deep reinforcement learning guided improvement heuristic for job shop scheduling. In _ICLR_.
>
> [13] Park, J., Chun, J., Kim, S. H., Kim, Y., \& Park, J. (2021). Learning to schedule job-shop problems: representation and policy learning using graph neural network and reinforcement learning. _International Journal of Production Research_.
>
> [14] Su, C., Zhang, C., Xia, D., Han, B., Wang, C., Chen, G., \& Xie, L. (2023). Evolution strategies-based optimized graph reinforcement learning for solving dynamic job shop scheduling problem. _Applied Soft Computing_.
>
> [15] Huang, J. P., Gao, L., \& Li, X. Y. (2024). An end-to-end deep reinforcement learning method based on graph neural network for distributed job-shop scheduling problem. _Expert Systems with Applications_.
>
> [16] Jayanetti, A., Halgamuge, S., \& Buyya, R. (2024). Reinforcement learning based workflow scheduling in cloud and edge computing environments: a taxonomy, review and future directions. arXiv:2408.02938.

---

> ### Author Response · Authors · 2024-11-20
> **Response to Reviewer 8vKM cnt.**
>
> **W2: Difference between our method and Wang et al. 2022 [7].**
>
> We will cite [7] and clarify our key differences from [7]:
>
> 1. **Problem Scope**: [7] focuses on failure-predictive maintenance scheduling, while we address DWS. Our method is also **applicable to related scheduling problems** such as Flexible Job Shop Scheduling (FJSS) [4]. Table R1 demonstrates the transferability of our method, which achieves **competitive results**.
> | FJSS Size   | MOR    | SPT    | FIFO   | MWKR   | DRL-G  | DRL-S  | Ours   |
> |-------------|--------|--------|--------|--------|--------|--------|--------|
> | 10×5        | 116.69 | 129.06 | 119.62 | 115.29 | 111.67 | 105.61 | 112.57 |
> | 20×5        | 217.17 | 229.89 | 216.13 | 216.98 | 211.22 | 207.50 | 202.38 |
> | 30×10       | 320.18 | 347.40 | 328.50 | 319.89 | 313.04 | 312.20 | 304.63 |
> | 40×10       | 425.19 | 443.30 | 427.22 | 425.70 | 416.18 | 415.15 | 395.70 |
>
>     **Table R1. Results on FJSS instance with different sizes.**
>
> 2. **Problem Scale**: [7] addresses relatively *small problems* (up to 96 aircraft), while we handle much larger scales with up to 600,000 tasks, approx. **3,000 times difference in scale**.
>
> 3. **Task Dependencies and Graph Representation**: [7] assumes *independent* tasks. DWS involves **intricate inter-task dependencies, workflow interactions**, and **task-machine heterogeneity**. Our task-specific and system-oriented graph representations can effectively capture these complexities.
>
> 4. **RL Algorithm**: [7] *avoids Actor-Critic* methods. In contrast, we propose specialized graph representation for the critic and incorporate self-attention layers to significantly **enhance value function accuracy** and **stabilize online training**.
>
> 5. **Online Learning**: Online learning in [7] refers to the adaptation of *predicted failures*. In contrast, our method **continuously improves** the offline-trained actor using real-world experiences, ensuring **adaptability to dynamic demands**.
>
> We will include these distinctions in the revised paper to highlight the novelty of our approach.
>
> **W3: Clarify the significance of improvements provided by the Online RL component.**
>
> The Online RL component is important in ensuring **long-term stability** and **practical utility** for **large-scale** DWS. It is valuable in two key aspects:
>
> 1. **Long-term utility**: Online RL continuously learns from new experiences, **avoiding the costly retraining required by offline RL**. It is highly sample-efficient, responsive, and adaptable, making it essential for maintaining high performance in dynamic environments.
>
> 2. **Practical significance**: While the 2% improvement may appear small, its absolute impact is substantial for large-scale problems. As shown in Table R2, even slight reductions in flowtime (e.g., 0.84%) can **lead to significant cost savings** (e.g., 36.11% in machine rental fees), demonstrating the **broader practical benefits** (paper will be updated accordingly).
>
> | Scenarios   | Objectives | Plan-1 | Plan-2 | Diff.           |
> |-------------|------------|----------|----------|-----------------|
> | ⟨5×5,5.4,30⟩ | Flowtime   | 421.94   | 420.29   | 0.39% |
> |             | Cost       | 102.44   | 82.28    | 19.68% |
> | ⟨5×5,9,30⟩  | Flowtime   | 416.53   | 413.02   | 0.84% |
> |             | Cost       | 152.63   | 97.51    | 36.11% |
> | ⟨6×4,5.4,30⟩ | Flowtime   | 292.81   | 286.73   | 2.08%  |
> |             | Cost       | 188.43   | 143.47   | 23.86%  |
> | ⟨6×4,9,30⟩  | Flowtime   | 309.70   | 306.90   | 0.90%  |
> |             | Cost       | 137.43   | 91.18    | 33.65%  |
>
> **Table R2. Comparison of two scheduling plans in flowtime and cost.**

---

> ### Author Response · Authors · 2024-11-20
> **Response to Reviewer 8vKM cnt.**
>
> **Q1: The difference from [7].**
>
> Please refer to the response of W2.
>
> **Q2: Accounting for agent-task heterogeneity in the graph representation.**
>
> We account for agent-task heterogeneity in the graph representation through the following:
>
> 1. **Raw Features**: Each task node includes static features (e.g., task workload) and **dynamic features** (e.g., execution time, machine utilization) that are **updated** at each decision step to capture task and machine heterogeneity (Appendix C).
>
> 2. **Edges**:
>    - Edges that model workflow-specific task execution order.
>    - Edges that model machine-specific task execution order.
>
> 3. **Dynamic Graph Structure**: The graph evolves its structure in real-time as tasks are completed, new tasks arrive, and machine states change in a dynamic environment.
>
> **Q3: Novelty of this work compared to prior research.**
>
> The novelty of our work lies in the following key aspects compared to existing methods [5-6]:
>
> 1. **Dynamic and evolving environments**: Our graph representation and architecture effectively handle **dynamic** workflow arrivals, **heterogeneous** machines, and **evolving** task dependencies, making it highly suited for cloud computing environments.
>
> 2. **Integrated offline and online learning**: We combine **offline and online RL** seamlessly, enabling _rapid adaptation to large-scale, dynamic problems without unnecessary retraining_.
>
> 3. **Scalability and extensibility**: Our method handles **extremely large** problems (up to 600,000 tasks) and achieves SOTA performance on related scheduling problems like FJSS [4]. New experiments to be included in the revised paper also demonstrate its ability to **support multi-objective** DWS (Please refer to response W4 to Reviewer RdYS).
>
> These innovations address **complex, large-scale, dynamic** scheduling problems rarely tackled in prior literature, advancing the _Learn-to-Optimize_ field.

---

> ### Comment · Reviewer_8vKM · 2024-11-26
>
> I would like to thank the authors for their detailed response to the questions posed, especially clarifying the challenges unique to their domain and detailed explanation of how they model their Graph Neural Network features. I have revised my scores in response.

---

> ### Author Response · Authors · 2024-11-26
>
> We greatly appreciate the reviewer for carefully reviewing our response,  recognizing the value of our work and raising the score!

---

### Official Review · Reviewer_Eeze · 2024-11-04

**Soundness:** 3
**Presentation:** 3
**Contribution:** 2
**Rating:** 5
**Confidence:** 3

**Summary:**

Prior works often consider homogeneous setups and static conditions, and fails to consider the dynamic nature of the workflow scheduling problem. To this end, the paper proposes a GNN-based DRL approach with an offline stage as well as an online stage.

**Strengths:**

- The paper has many diagrams, which help the readers to understand.
- GOODRL shows strong performance against other baselines.

**Weaknesses:**

- My main concern is that the paper seems to be applying some existing algorithms to a scheduling problem. There are some simple modifications at different parts of the overall method, and ended up giving us good performance. However, what are the broader insights of this work?
- Consider adding more background on the DWS problem and studies on real traces.

**Questions:**

1. In a real-world data cluster, how does the arrival pattern change over time? Consider plotting the 95%-99% quantile of the number of simultaneous arrivals. Please also consider adding other more background information.
2. How long does the model take to make a decision at inference time, compared to prior approaches?
3. It seems that in Table 1 and 2, the arrival rates during training and testing are always the same for each scenario. However, in a real-world data center, the arrival pattern might fluctuate over time, especially in the case of extreme events (e.g., holidays or deadlines). How robust is your approach to such distribution shifts?
4. In Table 1 and 2, "Ours-Offline" already achieves a very good performance. If due to distribution shifts, the offline version gets a much lower performance, can the online algorithm quickly adapt to such changes?
5. Many real-world scenarios involve some type of resource contention or performance interference. For example, two tasks are both memory intensive, so maybe we should allocate them on different machines. How does GOODRL address this issue?

Minor:
- Line 45, "In fact existing" --> "In fact, existing"

---

> ### Author Response · Authors · 2024-11-20
> **Response to Reviewer Eeze**
>
> We thank the reviewer for their thoughtful comments and will address these concerns in the revised paper.
>
> **W1: The broader insights with respect to simple algorithm modifications.**
>
> Our work addresses **large-scale dynamic** scheduling problems with **up to 600,000 tasks**, **heterogeneous** machines, and **unpredictable** workflow arrivals, demanding for major modifications to graph representations, network architectures, and training methods.
>
> By tackling such challenging scheduling problems, we derive broader insights in the following aspects：
> - The graph-based input to _the actor and critic networks should be clearly **separated**_ to effectively address large-scale, dynamic scheduling problems. This separation ensures a balance between global information capture and computational efficiency, particularly for the actor.
> - The actor and critic networks _must be designed to meet their **specific needs** in PPO_. The actor differentiates actions for the target task, while the critic focuses on global information. Therefore, they should be trained separately rather than jointly, as in previous works.
> - Actor-critic algorithms like PPO can _become **unstable** in large-scale, dynamic scheduling problems_. Its learning stability can be noticeably improved through gradient control and high-frequency independent training of the critic.
> - Our experiments in Table R1 indicate that GOODRL, while designed for DWS, also _performs competitively in other scheduling problems_ like FJSS [1], demonstrating its transferability.
> | Size   | MOR    | SPT    | FIFO   | MWKR   | DRL-G  | DRL-S  | Ours   |
> |-------------|--------|--------|--------|--------|--------|--------|--------|
> | 10×5        | 116.69 | 129.06 | 119.62 | 115.29 | 111.67 | **105.61** | 112.57 |
> | 20×5        | 217.17 | 229.89 | 216.13 | 216.98 | 211.22 | 207.50 | **202.38** |
> | 30×10       | 320.18 | 347.40 | 328.50 | 319.89 | 313.04 | 312.20 | **304.63** |
> | 40×10       | 425.19 | 443.30 | 427.22 | 425.70 | 416.18 | 415.15 | **395.70** |
>
>     **Table R1. Results on FJSS instances with different sizes.**
>
> **W2: Consider adding more background on the DWS problem and studies on real traces.**
>
> We will incorporate more background information on the DWS problem into the revised paper.
>
> For real-world traces, existing studies have explored three main aspects, which are also considered by us:
> - In line with many existing studies [2-5], we focused on popular **scientific workflows** like CyberShake, Montage, and SIPHT (https://download.pegasus.isi.edu/misc/SyntheticWorkflows.tar.gz) in our experiments;
> - We adopted **real-world resource configurations** from major cloud providers like Amazon EC2;
> - Following SOTA research [3-7], we **simulated workflow arrivals** using Poisson distributions under a wide range of arrival rates.
>
> We will update the manuscript to reflect this grounding in the literature and its relevance to practical scenarios.
>
> To the best of our knowledge, **no real-world workflow arrival patterns** are referenced or used in existing studies. If such traces exist, they have not been accessible to us. We welcome any suggestions from the reviewer regarding this.
>
> ---
> [1] Song, W., Chen, X., Li, Q., \& Cao, Z. (2022). Flexible job-shop scheduling via graph neural network and deep reinforcement learning. _IEEE Transactions on Industrial Informatics_.
>
> [2] Deelman, E., Vahi, K., Juve, G., Rynge, M., Callaghan, S., Maechling, P.J., Mayani, R., Chen, W., Da Silva, R.F., Livny, M. \& Wenger, K. (2015). Pegasus, a workflow management system for science automation. _Future Generation Computer Systems_.
>
> [3] Xie, Y., Wang, X. Y., Shen, Z. J., Sheng, Y. H., \& Wu, G. X. (2023). A two-stage estimation of distribution algorithm with heuristics for energy-aware cloud workflow scheduling. _IEEE Transactions on Services Computing_.
>
> [4] Qin, S., Pi, D., Shao, Z., Xu, Y., \& Chen, Y. (2023). Reliability-aware multi-objective memetic algorithm for workflow scheduling problem in multi-cloud system. _IEEE Transactions on Parallel and Distributed Systems_.
>
> [5] Sun, Z., Mei, Y., Zhang, F., Huang, H., Gu, C., \& Zhang, M. (2024). Multi-Tree Genetic Programming Hyper-Heuristic for Dynamic Flexible Workflow Scheduling in Multi-Clouds. _IEEE Transactions on Services Computing_.
>
> [6] Wang, S., Li, X., \& Ruiz, R. (2019). Performance analysis for heterogeneous cloud servers using queueing theory. _IEEE Transactions on Computers_.
>
> [7] Gu, C., Li, Z., Huang, H., \& Jia, X. (2018). Energy efficient scheduling of servers with multi-sleep modes for cloud data center. _IEEE Transactions on Cloud Computing_.

---

> ### Author Response · Authors · 2024-11-20
> **Response to Reviewer Eeze cnt.**
>
> **Q1: Temporal variability in workflow arrival patterns and their quantile plot.**
>
> Workflow scheduling research typically assumes **Poisson-distributed arrival patterns** [4-6], supported theoretically by [6]. While some studies [5,7] considered real-world data, the actual arrival times are still simulated by Poisson distributions. Our work aligns with this common practice in the literature, which will be further clarified in the revised paper. Furthermore, our online learning method can **adapt to substantial temporal variability in arrival rates or patterns**, meeting the critical demands of real-world applications.
>
> The 95\%-99\% quantiles of workflow arrivals at different arrival rates are presented in Table R2. The corresponding plots will be included in the revised paper.
> | Arrival Rates (workflows/h) | 95% Quantile           | 99% Quantile           |
> |-----------------------------|------------------------|------------------------|
> | $\lambda=5.4$               | 9.3 (approx. 280 tasks/h) | 11.7 (approx. 350 tasks/h) |
> | $\lambda = 9.0$             | 14.0 (approx. 420 tasks/h) | 16.0 (approx. 480 tasks/h) |
>
> **Table R2. 95\%-99\% quantiles of workflow arrivals at different arrival rates.**
>
> **Q2: Inference time efficiency compared to baseline approaches.**
>
> As shown in Table R3, our model takes **only 6-7 ms** to make a decision. Although ESRL and GPHH are faster, our model's inference time is less than the communication latency and data transfer time in cloud, hence **short enough to meet real-world requirements**. New results and discussions will be updated in the revised paper.
> | Scenarios                    | GPHH   | ESRL   | Ours   |
> |------------------------------|--------|--------|--------|
> | $\langle 5 \times 5,5.4,1k \rangle$ | 0.7 ms | 2.6 ms | 6.1 ms |
> | $\langle 5 \times 5,9,1k \rangle$   | 1.0 ms | 2.7 ms | 7.6 ms |
> | $\langle 6 \times 4,5.4,1k \rangle$ | 0.6 ms | 2.7 ms | 6 ms   |
> | $\langle 6 \times 4,9,1k \rangle$   | 0.7 ms | 2.5 ms | 6.8 ms |
>
> **Table R3. The inference time to make a decision.**
>
> **Q3: Robustness of the approach to fluctuating workflow arrivals.**
>
> We agree that workflow arrivals can fluctuate during extreme events. We conducted further experiments where we **varied the arrival rate** by ±50% to +100% compared to the training data. In Table R4, GOODRL **consistently outperforms all** competing approaches, demonstrating its **robustness to such fluctuations** and ability to adapt to dynamic changes.
> | Arrival Rates | EST     | PEFT   | HEFT   | GP     | ESRL    | Ours   |
> |---------------|---------|--------|--------|--------|---------|--------|
> | -50%          | 1288.59 | 626.37 | 567.55 | 403.99 | 2328.24 | **398.01** |
> | +50%          | 1165.15 | 516.60 | 481.70 | 413.99 | 4076.43 | **409.66** |
> | +100%         | 1112.05 | 498.06 | 469.46 | 424.92 | 5255.60 | **423.02** |
>
> **Table R4. Performance comparison under changed arrival rates.** (to be included in the revised paper)
>
> **Q4: Adaptability of the online algorithm to performance drops of the offline-trained actor.**
>
> Shifts in workflow arrival rate can impact the performance of the offline-trained actor. However, the mean flowtime **remains robust upon increasing the arrival rate** from 5.4 to 9 (see Table 2 in the paper). To further investigate, we conducted **additional experiments** to test the adaptability of our online algorithm.
>
> We introduced extra noise $\epsilon$ at different levels to degrade the offline actor’s performance and trained this noise-infused actor online. In Table R5, performance initially dropped by 3.7–3.8\%, but after 150 online training iterations, the **gap reduced** to 1–2\%. Hence, online learning can **quickly adapt to distribution shifts** and **recover performance** effectively.
>
> | Noise       | 0-th   | 25-th  | 50-th  | 75-th  | 100-th | 125-th | 150-th |
> |-------------|--------|--------|--------|--------|--------|--------|--------|
> | ε = 0.05    | 3.71%  | 3.61%  | 3.49%  | 3.20%  | 2.21%  | 1.94%  | 1.95%  |
> | ε = 0.1     | 3.87%  | 3.66%  | 2.92%  | 2.42%  | 1.46%  | 1.10%  | 1.06%  |
>
> **Table R5. Performance gap with the no-nosie actor across online training iterations.** (to be included in the revised paper)
>
> **Q5: Handling resource contention and performance interference in task allocation**
>
> Resource contention typically arises when multiple tasks run simultaneously on the same VM. In dynamic workflow scheduling, each VM **processes its tasks sequentially**. We also ensure that each VM has sufficient memory before assigning a task. Thus, **resource contention is unlikely to be a significant issue** in our problem. We will clarify this in the revised paper.

---

> ### Author Response · Authors · 2024-11-28
> **Kindly request feedback from Reviewer Eeze**
>
> Dear Reviewer Eeze,
>
> We sincerely thank you for your time and effort in reviewing our work. As the discussion phase soon approaches to its end, we would greatly appreciate it if you could kindly provide feedback on our responses. We are eager to engage in further discussion with you and are open to new suggestions.
>
> We would like to provide more explanations to address your concerns regarding **our work's differences from existing methods** and the **problem background**, with all the relevant changes _highlighted in red_ in the revised paper. Key explanations are summarized below for your convenience.
>
> 1. To address the unique challenges posed by **large-scale dynamic** workflow scheduling (DWS) problems, our work introduces **three significant modifications** that set it apart from existing studies, including:
>
>     - **Graph representations**: Many existing scheduling methods were designed to solve *small-scale* problems with *fixed graph structures* (i.e., the number of nodes in the graph-based state representation remains fixed). In contrast, our approach utilizes **dynamic graph representations** to effectively capture the **time-varying relationships** among completed, ongoing, and newly arrived workflows, while simultaneously tracking **real-time machine status**. This design ensures a comprehensive and up-to-date view of the scheduling environment.
>     - **Network architectures**: In most prior studies, the actor and critic share the *same feature extraction layers* and rely on the *same state input*. Instead, we propose a new actor-critic architecture that allows the actor and the critic to **process different state representations**. In this way, the actor is tailored to distinguishing important actions. The critic focuses on processing the state information at the global scale. The effectiveness of this new architecture and its advantageous over other competing approaches have been **verified experimentally** on a range of large and dynamic scheduling scenarios.
>     - **Training methods**: Unlike many previous works that *apply existing RL algorithms without problem-specific modifications*, we propose a **novel offline-online learning** method to achieve reliable online improvement of the actor during the daily operation of the scheduler, significantly enhancing the actor's adaptability and performance on large and dynamic scheduling problems.
>
> 2. Following your advice, we further revised our paper to **strengthen the background discussion** regarding the DWS problem. We clearly **highlighted the key focus** of DWS, which is to assign a long sequence of interdependent tasks to heterogeneous virtual machines, driven by the aim to optimize the _mean flowtime_ across all workflows or other objectives such as the cost (see Appendix O of the revised paper for new experiment results).
> There are some **key assumptions** associated with the DWS problems, including:
>     - The pattern of each workflow (i.e., DAG structure) is unknown until it reaches the system.
>     - Tasks within a workflow can be allocated to any machine, with processing times varying according to machine speeds.
>     - Each machine can process only one task at a time.
>     - Only tasks with all predecessors completed are eligible for scheduling.
>
>     All these assumptions **match closely** with numerous real-world applications. We have provided more background information in Appendix B and I of the revised paper.
>
> Best regards,
> Paper 4258 Authors

---

> ### Author Response · Authors · 2024-12-02
> **Kindly ask for feedback from Reviewer Eeze**
>
> Dear Reviewer Eeze,
>
> As the discussion phase is nearing its conclusion, we kindly ask if you could provide feedback on our responses and revisions. If you find that we have satisfactorily addressed your concerns, we would greatly appreciate your consideration of adjusting your rating to reflect the improvements made.
>
> We sincerely thank you for your time and thoughtful review.
>
> Best regards,
> Paper 4258 Authors

---

> > ### Comment · Reviewer_Eeze · 2024-12-02
> >
> > Dear authors,
> >
> > Thank you for your detailed response and the comprehensive experiments that are added. I have thoroughly reviewed your updated manuscript, as well as your responses to the other reviewers. I don't have any further questions on how well your approach works on this particular problem --- your experiments demonstrate this part very well. However, the three contributions you highlighted, 1) applying GAT to each task structure, 2) using separate encoders for actor and critic, and 3) offline imitation learning and online adaptation, they don't seem to be entirely novel for a ML conference. To me, they seem more like taking off-the-shelf methods, and tailoring them for a specific system problem. For this reason, while I appreciate the comprehensiveness of your experiments and the clarity of your writing, I think this paper might be more appropriate for a systems conference.

---

> > > ### Author Response · Authors · 2024-12-03
> > > **Further Response to Reviewer Eeze - [1/2]**
> > >
> > > Dear Reviewer Eeze,
> > >
> > > Thank you for reviewing our updated manuscript and recognizing the comprehensiveness of our experiments and the clarity of our writing. We truly appreciate your feedback.
> > >
> > > We would like to provide response to your concerns about the applicability of our scheduling problems, the relevance to ML-focused conferences, the novelty of our technical contributions, and its relevance to the ICLR community.
> > >
> > > **1. The scheduling problem is widely applicable**
> > > Scheduling is a fundamental combinatorial optimization problem (COP) with **wide-ranging applications**, including but not limited to cloud computing, manufacturing, maritime, aviation, and logistics. **Its importance is well-recognized in the ML community**, as evidenced by prior works published at the **top-tier ML conferences**:
> > > |   | **Influences** | **Graph Representations** | **Architectures** | **Training Methods** | **Problem Scales** |
> > > |-|-|-|-|-|-|
> > > | **[8] (NeurIPS 2020)** | Has been **cited 378 times** | Static disjunctive graphs | GIN; Shared encoder | Unmodified PPO | ≤2,000 tasks |
> > > | **[9] (ICLR 2024)** | **Highly rated** work (8,8,8,6 ratings) | Same as [8] | [8] + GAT; Only actor encoder | Unmodified REINFORCE | ≤2,000 tasks |
> > > | **[10] (NeurIPS 2024)** | **Quickly cited** by 9 papers | Same as [8] | [9] + attention layers; Only actor encoder | Self-supervised learning | ≤2,000 tasks |
> > > | **Our Work** | ---- | **New dynamic** graph | GAT+ attention layers; **Separate** encoders | **New** offline-online PPO | ≤**600,000** tasks |
> > >
> > > **Table R6. Summary of prior works in Learning-to-Optimize for scheduling problems.**
> > >
> > > Different from the three successful studies summarized above, **our work introduces innovations in all aspects** of graph representation, architectures, training methods, and problem scale. Additionally, our work clearly demonstrates the **wide applicability** of GOODRL, extending beyond DWS to FJSS (see our response W1 to Reviewer jCq6) as well as multi-objective problems (e.g., joint optimization of mean flowtime and VM rental cost, see our response W4 to Reviewer RdYS), showcasing **strong generalization capability**. This aligns closely with the **growing trend in the ML community** of applying deep learning techniques to increasingly large and dynamic scheduling problems, advancing **both methodology and real-world applications**.
> > >
> > > **2. ML research demands for both new techniques and tailored applications**
> > > We acknowledge that many _**Learning-to-Optimize**_ (**L2O**) studies, such as [8], [9], and [10], successfully **leveraged existing ML techniques** like GIN, GAT, and PPO **in innovative ways**. While these techniques are not entirely new to the broader ML field, _their adaptation to specific scheduling problems **provides valuable insights** and **lays the groundwork** for advancing ML applications_. Similarly, our approach makes a **significant contribution to the L2O field** by creatively leveraging and enhancing existing ML techniques (see point 3 below) to tackle **significantly larger** and **highly dynamic** scheduling problems with substantial practical importance.
> > > In addition, the **ICLR website** (https://iclr.cc/) mentions that ICLR is globally renowned for presenting and publishing cutting-edge research on all aspects of DL used in the fields of AI, statistics, and data science, **as well as important application** areas. The **relevant topics** discussed at the conference **also include applications** in fields such as audio, speech, robotics, neuroscience, biology, and others.

---

> ### Author Response · Authors · 2024-12-03
> **Further Response to Reviewer Eeze - [2/2]**
>
> **3. Reaffirming the novelty and significance of our contributions**
> Similar to other L2O works published in the top-tier ML conferences, while our work builds on existing ML techniques such as GAT and RL, it makes **new contributions to the ML community**, particularly in the L2O domain:
> | **Challenges**  | **Existing Limitations** | **Our Newly Proposed Techniques** |
> |-|-|-|
> | **Capturing changes in dynamic environments** | 1) Rely on a **static graph**, i.e., the number of nodes in graphs is constant as all information is known. 2) **Unable** to capture complex and dynamic relationships between workflows and machines. | **Dynamic graph representation with GAT for task structures.** To our knowledge, this type of dynamic modeling **has not been explored** in prior L2O works for scheduling. |
> | **Solving RL stability for large-scale problems** | 1) Use a **shared** or **only** an actor encoder. 2) **Neglecting** the critic's role in AC-based RL stability for large-scale problems (see our response Q1 to Reviewer jCq6 for details). | **Separate encoders for actor and critic** can significantly **enhance learning reliability and scalability** for solving COPs, including but not limited to scheduling. |
> | **Adapting to unpredictable future changes** | 1) Perform **offline learning** using existing methods. 2) **Unable** to continuously learn in the face of future environmental changes. | **Offline imitation learning with online adaptation.** To our knowledge, this framework **has not been applied** in the L2O domain, providing a novel and valuable contribution to L2O research. |
>
> **Table R7. Limitations of existing works and our novelties.**
>
> **4. Relevance to the ICLR community**
> As the **ICLR official account highlighted last week** (https://x.com/iclr_conf), ``_Does this paper take a gradient step in a promising direction? Is the community better off with this paper published? If the answer is yes, then the recommendation should be to accept_.’’
> - **X1：Yes**, our research takes **a positive step in a promising direction** by addressing challenging Dynamic Workflow Scheduling (DWS) problems through the introduction of dynamic graph representations, separately designed and trained actor and critic networks, and a hybrid offline-online learning framework. These innovations enhance scalability, stability, and adaptability, **pushing the boundaries of L2O methods** and **paving the way for broader real-world applications**.
> - **X2: Yes**, publishing this paper will **greatly benefit the ICLR community** by introducing innovative ML techniques that are not only applicable to DWS but **also transferable to other frequently studied problems**, such as FJSS (see Appendix N) and multi-objective optimization (see Appendix O). Our in-depth study provides valuable insights into reliable online learning methods, advanced state representations for dynamic environments, and novel actor-critic network architectures (see Appendix F and G) that **scale effectively to large problem sizes**. These contributions establish **a solid foundation** for advancing both algorithmic research and practical applications in the L2O field.
>
> We hope this response clarifies the broader relevance and novelty of our work. Would you please reconsider your evaluation in light of this discussion? Thank you again for your constructive feedback and consideration.
>
> Best regards,
> Paper 4258 Authors
>
> ---
> [8] Zhang, C., Song, W., Cao, Z., Zhang, J., Tan, P. S., & Chi, X. (2020). Learning to dispatch for job shop scheduling via deep reinforcement learning. In _NeurIPS_.
>
> [9] Zhang, C., Cao, Z., Song, W., Wu, Y., & Zhang, J. (2024). Deep reinforcement learning guided improvement heuristic for job shop scheduling. In _ICLR_.
>
> [10] Corsini, A., Porrello, A., Calderara, S., & Dell'Amico, M. (2024). Self-labeling the job shop scheduling problem. In _NeurIPS_.

---

### Official Review · Reviewer_jCq6 · 2024-11-05

**Soundness:** 2
**Presentation:** 3
**Contribution:** 3
**Rating:** 8
**Confidence:** 3

**Summary:**

This paper proposes a novel Graph-assisted Offline-Online Deep Reinforcement Learning (GOODRL) approach for Dynamic Workflow Scheduling (DWS) in cloud environments. The authors introduces three main innovations: a task-specific graph representation with a Graph Attention Actor Network for focused task assignments, a system-oriented graph representation with a Graph Attention Critic Network for real-time state evaluation, and a hybrid offline-online RL method to improve adaptability. The offline stage uses imitation learning for stable initial policy development, while the online stage applies advanced PPO techniques for continuous adaptation. Experiments demonstrate GOODRL’s superiority over state-of-the-art methods in minimizing mean flowtime and enhancing performance in several offline and online settings.

**Strengths:**

1. The paper is well-written and easy to follow
2. Special designs on the actor and critic network to have more efficient embeddings and long-range interaction modeling
3. customized gradient for stabilizing the PPO training for the online settings
4. Experiments are conducted on many online and offline settings

**Weaknesses:**

This paper presents a comprehensive learning pipeline for addressing the dynamic workflow scheduling (DWS) problem. I appreciate the authors for their efforts in adapting various components to suit the unique challenges of DWS.

The primary concern with this paper is the applicability of the proposed pipeline. Many of the modifications and design choices appear closely tailored to DWS, leaving it unclear how generalizable this approach might be to other scheduling problems, such as flexible job shop scheduling. Can these designs be readily adapted for other problem domains? The paper would be significantly strengthened by demonstrating the pipeline’s transferability to other scheduling scenarios.

Several techniques are introduced throughout the pipeline, though not all are rigorously validated in the ablation study. A more thorough investigation into the contributions of each component would enhance our understanding of their individual benefits.

Overall, I appreciate the contributions of this work and currently lean toward a borderline acceptance.

---
I increase my score to 8 in response to the author's rebuttal.

**Questions:**

Besides the concerns raised in the weakness part, I have the following additional questions:

1. PPO has also been applied to solve other combinatorial optimization problems like routing problems, where the horizons are also very large. Could you give some intuitions why PPO is particularly unstable for this problem?

---

> ### Author Response · Authors · 2024-11-20
> **Response to Reviewer jCq6**
>
> We sincerely thank your positive feedback and valuable suggestions on our work. We hope our following response can address your concerns. All additional experiments and discussions will be included in the revised paper soon.
>
> **W1: Can these designs be readily adapted for other problem domains, such as flexible job shop scheduling?**
>
> We confirm that our designs **can be readily adapted** for other scheduling problems, such as Flexible Job Shop Scheduling (FJSS) problems. To address your concern, we conducted **additional experiments** on FJSS from [1], as it is a strong representative and was mentioned by (Reviewer 8vKM). Table R1 showd that our method achieved highly **competitive** performance, demonstrating the **transferability** of our pipeline (i.e., GOODRL).
>
> | Size   | MOR    | SPT    | FIFO   | MWKR   | DRL-G  | DRL-S  | Ours   |
> |-------------|--------|--------|--------|--------|--------|--------|--------|
> | 10×5        | 116.69 | 129.06 | 119.62 | 115.29 | 111.67 | **105.61** | 112.57 |
> | 20×5        | 217.17 | 229.89 | 216.13 | 216.98 | 211.22 | 207.50 | **202.38** |
> | 30×10       | 320.18 | 347.40 | 328.50 | 319.89 | 313.04 | 312.20 | **304.63** |
> | 40×10       | 425.19 | 443.30 | 427.22 | 425.70 | 416.18 | 415.15 | **395.70** |
>
> **Table R1. Results on FJSS instance with different size.**
>
> In our experiments, all sequence-structured jobs within a FJSS instance are _represented jointly as a workflow_, enabling our pipeline to handle FJSS effectively as a special case of our workflow scheduling problem.
>
> **W2: Several techniques are introduced in the pipeline, but not all are rigorously validated in the ablation study.**
>
> To address your concern, we conducted comprehensive ablation studies to validate the effectiveness of the key techniques introduced in our pipeline. Ablation experiments cover **three aspects**:
>
> 1. **Actor Network**: Table R2 validated the **task-specific embedding module** (TSEM) in our actor network introduced in Subsection 4.2.1. Our architecture with pairwise processing and focused task embedding (**Ours-TSEM**) outperformed the baselines (TSEM w/o. pair and TSEM w. mean), achieving the **lowest** cross-entropy loss. This highlights the importance of _separating task-machine pairs_ and avoiding mean pooling, adequately prioritizing critical task-specific information for decision-making.
> | Actor Architecture | 100-th  | 200-th  | 300-th  | 400-th  | 500-th  |
> |--------------------|---------|---------|---------|---------|---------|
> | Ours-TSEM          | 2.7486  | **2.7106**  | **2.6881**  | **2.6647**  | **2.6498**  |
> | TSEM w/o. pair     | 3.1707  | 3.1597  | 3.1538  | 3.1468  | 3.1435  |
> | TSEM w. mean       | **2.7099**  | 2.7209  | 2.7152  | 2.6659  | 2.7109  |
>
>     **Table R2. Cross-entropy loss at different iterations.**
>
> 2. **Critic Network**: Table R3 validated the **system-oriented embedding module** (SOEM) in our critic network introduced in Subsection 4.2.2. **Our-SOME** clearly **outperformed** variants like SOEM w/o. edge (remove bi-directional and additional task-machine edges) and SOEM w/o. self (remove self-attention layers) in value loss. These results highlight the importance of _comprehensive context awareness_ and _long-range interaction_ modeling developed by us.
> | Critic Architecture | 100-th    | 200-th    | 300-th    | 400-th    | 500-th   |
> |---------------------|-----------|-----------|-----------|-----------|-----------|
> | Ours-SOEM           | **16.3971** | 14.0938   | **10.4907** | **9.5811**  | **7.8581**   |
> | SOEM w/o. edge      | 17.3012   | **13.4737** | 11.6626   | 9.8066    | 8.8853    |
> | SOEM w/o. self      | 20.6114   | 16.1826   | 14.6813   | 12.6997   | 12.0733   |
>
>     **Table R3. Mean relative error between return and state value at different iterations.**
>
> 3. **Online Learning**: Table R4 validated two techniques for online learning introduced in Subsection 4.3.2. **Ours-Online** achieved **superior** online performance improvement compared to Online w/o. grad. (remove gradient control) and Online w/o. freq. (remove high-frequency critic updates). These results demonstrate the effectiveness of both techniques in stabilizing and enhancing online learning.
> | Training Method   | 150-th  | 175-th  | 200-th  | 225-th  | 250-th  |
> |-------------------|---------|---------|---------|---------|---------|
> | Ours-Online       | **1.62%** | **1.50%** | **1.57%** | **1.52%** | **1.52%** |
> | Online w/o. grad. | -1.18%  | -1.08%  | -1.24%  | -1.36%  | -1.64%  |
> | Online w/o. freq. | -184.80%| -261.27%| -283.93%| -336.86%| -382.54%|
>
>     **Table R4. Improvement in mean flowtime compared to Ours-Offline.**
>
> We believe our ablations studies (see Appendices F, G and L) provide rigorous validation suggested by the reviewer.
>
> ---
> [1] Song, W., et al. (2022). Flexible job-shop scheduling via graph neural network and deep reinforcement learning. _IEEE Transactions on Industrial Informatics_, 19(2), 1600-1610.

---

> ### Author Response · Authors · 2024-11-20
> **Response to Reviewer jCq6 cnt.**
>
> **Q1: Explain the instability of PPO, in view of its application to long-horizon routing problems.**
>
> The instability of PPO in our study is attributed to the unique challenges in horizon length, problem dynamicity, and complexity.
>
> 1. **Extremely Long Horizon**: While routing problems in [2] and other recent works [3-4] have long horizons of up to $7\times10^3$ decision steps, our DWS problem involves scheduling up to **$6\times10^5$** workflow tasks (approx. **100 times longer** in horizon length). Such long horizon has been shown to seriously affect PPO's instability in [5].
>
> 2. **Dynamic and Evolving Environments**: In DWS, **workflows arrive unpredictably with random patterns**. The system state evolves as tasks are completed and new workflows arrive. Such **environmental dynamicity** may seriously **interfere with PPO’s training process**, which assumes more predictable state transitions [6-7].
>
> 3. **Complex Task-Machine Dependencies**: The time of processing a task on different machines can **vary 12 times**. DWS also involves **more flexible machine selections** and **intricate task-machine dependencies**, requiring precise value approximations and robust policy updates. These complexities challenge PPO’s ability to maintain stability during training.
>
> These factors collectively explain why PPO experiences stability issues in our study. We will include additional discussions in the revised paper.
>
> ---
> [2] Hou, Q., Yang, J., Su, Y., Wang, X., \& Deng, Y. (2023). Generalize learned heuristics to solve large-scale vehicle routing problems in real-time. In _ICLR_.
>
> [3] Zhou, J., Cao, Z., Wu, Y., Song, W., Ma, Y., Zhang, J., \& Xu, C. (2024). MVMoE: Multi-task vehicle routing solver with mixture-of-experts. In _ICML_.
>
> [4] Bi, J., Ma, Y., Zhou, J., Song, W., Cao, Z., Wu, Y., \& Zhang, J. (2024). Learning to handle complex constraints for vehicle routing problems. In _NeurIPS_.
>
> [5] Queeney, J., Paschalidis, Y., \& Cassandras, C. G. (2021). Generalized proximal policy optimization with sample reuse. In _NeurIPS_.
>
> [6] Pan, F., Cai, Q., Zeng, A.X., Pan, C.X., Da, Q., He, H., He, Q., \& Tang, P. (2019). Policy optimization with model-based explorations. In _AAAI_.
>
> [7] Liu, S. (2024). An evaluation of DDPG, TD3, SAC, and PPO: Deep reinforcement learning algorithms for controlling continuous system. In _DAI_.

---

> > ### Comment · Reviewer_jCq6 · 2024-11-27
> >
> > Well done on the rebuttal! My concerns have been well-addressed. This paper has been strengthened with all these additional experiments. I believe this paper is above "borderline acceptance" and deserves an "acceptance". However, ICLR does not offer the choice of a score of 7. For now, I increase my score to 8.

---

> > > ### Author Response · Authors · 2024-11-27
> > >
> > > We deeply thank Reviewer jCq6 for the encouraging feedback, recognizing our efforts in the rebuttal, and raising the score to 8! Your comments truly motivate us, and we sincerely appreciate your recommendation.

---

### Author Response · Authors · 2024-11-21
**General Response**

We sincerely thank the reviewers for their thorough evaluation and thoughtful feedback. We are delighted that the reviewers recognized the importance of our research topic in cloud computing (jCq6, 8vKM), the technical novelty and effectiveness of our approach (jCq6, RdYS), the strong performance compared to baselines (jCq6, Eeze, 8vKM, RdYS), the comprehensive nature of our experiments (jCq6, RdYS), and the clarity and quality of our presentation (jCq6, Eeze, 8vKM, RdYS).

The insightful comments have helped us refine our work. Detailed responses to each question are provided below, along with new experiments, additional discussions, and updated background information, which will be incorporated into the revised paper and appendix. We believe we have addressed all key concerns raised and kindly encourage reviewers to review our rebuttals and provide prompt feedback at their earliest convenience. We are happy to engage further with reviewers to clarify any remaining issues or explore additional suggestions.

---

> ### Author Response · Authors · 2024-11-22
> **General Response Updated**
>
> Dear Reviewers,
>
> We are pleased to inform you that a revised version of our paper has been uploaded to address all your concerns. All modifications have been **highlighted in red** for your convenience. We have made every effort to incorporate your suggestions to the fullest extent possible. We believe these updates satisfactorily resolve the issues outlined, aligning closely with the responses provided in our rebuttals.
>
> We greatly value your feedback and would appreciate any further suggestions for improvement. If you find that our revisions adequately address your concerns, we kindly ask you to consider adjusting your ratings to reflect the updated version of our paper.
>
> Thank you very much for your time and thoughtful evaluation of our work. We remain open to further clarifications or refinements as requested.
>
> The Authors

---

### Meta-Review · Area_Chair_z3Ps · 2024-12-21

**Metareview:**

This paper presents an approach for dynamic workflow scheduling in cloud environments. The method combines task-specific graph representations with graph attention networks for actor-critic networks and integrates offline imitation learning with online reinforcement learning. The reviewers acknowledged the paper's strong empirical results, clear presentation, and practical value. While initial concerns were raised about technical novelty, experiments across varied workloads, and computational overhead, the authors provided comprehensive responses including additional experiments on multi-objective optimization, scalability analysis, and detailed comparisons with existing approaches.

**Additional Comments On Reviewer Discussion:**

Initially, key concerns centered on the method's novelty compared to existing scheduling approaches, its practical applicability, and computational requirements. The authors provided responses including new experimental results. This led to constructive dialogue with some reviewers explicitly raising their scores based on the thorough responses.

---

### Decision · Program_Chairs · 2025-01-22

Accept (Poster)